# Ergodic Risk Measures: Towards a Risk-Aware Foundation for Continual Reinforcement Learning

## Abstract

Continual reinforcement learning (continual RL) seeks to formalize the notions of lifelong learning and endless adaptation in RL. In particular, the aim of continual RL is to develop RL agents that can maintain a careful balance between retaining useful information and adapting to new situations. To date, continual RL has been explored almost exclusively through the lens of risk-neutral decision-making, in which the agent aims to optimize the expected long-run performance. In this work, we present the first formal theoretical treatment of continual RL through the lens of *risk-aware* decision-making, in which the behaviour of the agent is directed towards optimizing a measure of long-run performance beyond the mean. In particular, we show that the classical theory of risk measures, widely used as a theoretical foundation in non-continual risk-aware RL, is, in its current form, incompatible with continual learning. Then, building on this insight, we extend risk measure theory into the continual setting by introducing a new class of *ergodic* risk measures that are compatible with continual learning. Finally, we provide a case study of risk-aware continual learning, along with empirical results, which show the intuitive appeal of ergodic risk measures in continual settings.

## 1 Introduction

Reinforcement learning (RL) (Sutton and Barto, 2018) has enjoyed success over the years when tackling certain problems of interest in various domains, ranging from video games to robotics. However, the RL agents behind these successes are typically trained in static environments, and are evaluated on a single task for a finite period of time. By contrast, RL agents deployed in the real world may be required to operate indefinitely in scenarios where the environment and/or task changes over time. This discrepancy has motivated the study of continual reinforcement learning (continual RL) (Ring, 1994; Khetarpal et al., 2022; Abel et al., 2023; Kumar et al., 2025), which formalizes the challenges of lifelong learning and endless adaptation in RL. At the heart of continual RL is the *stability-plasticity dilemma*, through which an agent must learn to preserve sufficient prior knowledge, while still remaining sufficiently flexible to adapt to new streams of experience.

To date, the notion of continual RL has been studied almost exclusively under a *risk-neutral* lens, such that the behaviour of the agent is directed towards optimizing some expectation-based measure of long-run performance (e.g. the average per-step reward or the expected return). In this work, we present the first formal theoretical exploration of continual RL through the lens of *risk-aware* decision-making, in which the behaviour of the agent is directed towards optimizing a measure of long-run performance beyond the mean. Through this exploration, we seek to provide a theoretically-grounded formalism through which we can frame the notion of risk-awareness in a continual setting. Namely, the first half of our exploration is guided by the following questions:

▷ *What does it mean for an agent to be risk-aware in a continual setting?*

▷ *What are the conditions needed to enable an agent to be risk-aware in a continual setting?*

▷ *What are the implications of risk-awareness as it relates to the stability-plasticity dilemma?*

To answer these questions, we propose a risk-aware generalization of the continual RL framework proposed in Abel et al. (2025), where we formalize risk-aware continual RL as an ongoing exchange of actions and observations between an agent and an environment, such that the agent receives observations from the environment, processes those observations into *risk-aware observations* based on its *risk attitude* (or *risk tolerance*), then emits actions back to the environment based on the risk-aware observations. We then leverage this risk-aware framework to propose a set of axioms that establish the conditions needed to enable an agent to be risk-aware in a continual setting.

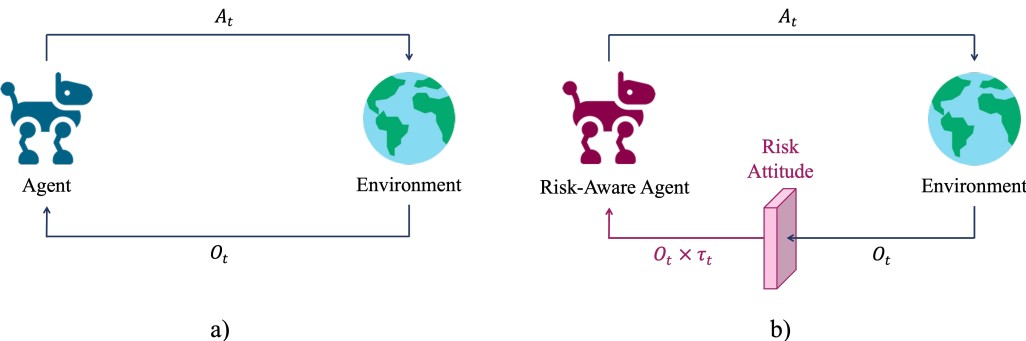

a)                                                            b)

Figure 1: A comparison of: **a)** the continual RL framework proposed in Abel et al. (2025), and **b)** the risk-aware generalization of the framework proposed in this work.

The second half of our exploration is then aimed at examining existing frameworks and methodologies used for non-continual risk-aware RL to see how compatible they are with our proposed framework. To this end, we first examine the classical theory of *risk measures* (e.g. see Chapter 6 of Shapiro et al. (2009)), which has served as a theoretical foundation for risk-aware decision-making in non-continual RL, and show that, in its current form, it is incompatible with continual learning. Then, building on this insight, we extend risk measure theory into the continual setting by proposing a new class of *ergodic* risk measures that are compatible with continual learning.

Finally, using the well-known average-reward Markov decision process (MDP) formulation (Puterman, 1994) as a basis, we provide a case study, along with numerical results, which show the intuitive appeal of ergodic risk measures in continual settings. Altogether, these contributions provide, to the best of our knowledge, the first formal theoretical treatment of risk-aware decision-making in a continual (i.e., lifelong) learning setting.

## 2 RELATED WORK

### 2.1 CONTINUAL REINFORCEMENT LEARNING

The notions of lifelong learning and endless adaptation in the context of RL have long been studied under different names and perspectives. In recent years, several works (e.g. Khetarpal et al. (2022); Abel et al. (2023); Kumar et al. (2025)) have attempted to unify and frame these diverse sets of works as instances of continual RL. Some of the more common types of RL-related works that can be interpreted as being instances of continual RL include the study of the loss of plasticity in deep RL agents (e.g. Abbas et al. (2023); Dohare et al. (2024)), transfer learning (e.g. Abel et al. (2018); Gimelfarb et al. (2021)), and decision-making in non-stationary environments (e.g. Dick et al. (2014); Luketina et al. (2022)). In this work, we place a great emphasis on the notion of *plasticity*, which has been studied extensively in prior works such as Raghavan and Balaprakash (2021), Chen et al. (2023), and Abel et al. (2025). In our proposed framework, we build on the definition of plasticity (and subsequent continual RL framework) proposed in Abel et al. (2025).

### 2.2 RISK-AWARE REINFORCEMENT LEARNING

The notion of risk-aware learning and decision-making in the context of RL has been studied under various theoretical frameworks, from the well-established expected utility framework (Howard and

Matheson, 1972), to the more contemporary framework of risk measures (e.g. Chapter 6 of Shapiro et al. (2009)). In this work, we focus on the latter framework, which originated in the finance literature (e.g. Rockafellar and Uryasev (2000)), but has since been widely integrated into RL-based works (e.g. Bäuerle and Ott (2011)). Traditionally, non-continual risk-aware RL works have aimed to optimize either a *static* risk measure (e.g. Mead et al. (2025)), or a *nested (dynamic)* risk measure (e.g. Wang and Delage (2025)). The case study presented in this work primarily focuses on the conditional value-at-risk (CVaR) risk measure (Rockafellar and Uryasev, 2000), which has been studied extensively in the discounted setting (e.g. Bäuerle and Ott (2011); Mead et al. (2025)), and, to a lesser extent, in the average-reward setting (e.g. Xia et al. (2023); Rojas and Lee (2025)).

## 3 PRELIMINARIES

### 3.1 CONTINUAL REINFORCEMENT LEARNING

Continual reinforcement learning (continual RL) can be viewed as an agent-environment interaction occurring over an infinite time horizon. In particular, following the framework proposed in Abel et al. (2025), this interaction can be modeled as an exchange of signals between an agent and an environment, where, at each discrete time step, $t$, the agent emits an action from an action-space, $A_t \in \mathcal{A}$, and the environment emits an observation from an observation-space, $O_t \in \mathcal{O}$ (see Figure 1a)). Accordingly, for a given agent and environment, we can define their interaction as a pair of sequences of random variables, $O_{a:b} \doteq (O_a, O_{a+1}, \ldots, O_{b-1}, O_b)$ and $A_{c:d} \doteq (A_c, A_{c+1}, \ldots, A_{d-1}, A_d)$, where $a, b, c,$ and $d$ denote discrete time indices.

One of the defining aspects of continual RL is the stability-plasticity dilemma, through which an agent must carefully balance the degree to which newly acquired information affects its behaviour relative to previously learned knowledge. In particular, the plasticity, $\mathfrak{P}$, of a given agent relative to a given environment with respect to the time intervals $[a:b] \doteq [a, a+1, \ldots, b-1, b], 1 \le a \le b$, and $[c:d] \doteq [c, c+1, \ldots, d-1, d], 1 \le c \le d$, can be defined as follows (Abel et al., 2025):

$$\mathfrak{P}_{\substack{a:b \\ c:d}} \doteq \mathbb{I}(O_{a:b} \to A_{c:d}), \tag{1}$$

where $\mathbb{I}(X_{a:b} \to Y_{c:d}) \doteq \sum_{i=\max(a,c)}^{d} \mathbb{I}(X_{a:\min(b,i)}; Y_i \mid X_{1:a-1}, Y_{1:i-1})$ denotes the *generalized directed information* (Abel et al., 2025) between two sequences of random variables, $X_{a:b}$ and $Y_{c:d}$, and $\mathbb{I}(X; Y)$ denotes the mutual information between random variables $X$ and $Y$. In words, the above definition interprets plasticity as a measure of how much a sequence of observations from the environment influences a sequence of the agent's actions. Accordingly, the stability-plasticity dilemma can be viewed as the dilemma associated with determining *the right amount of plasticity*.

### 3.2 RISK MEASURES

Let $(\Omega, \mathcal{F}, \mathbb{P})$ denote a probability space, and let $\mathcal{X}$ denote a space of random variables of the form $X : \Omega \to \mathbb{R}$. A *risk measure* (e.g. Chapter 6 of Shapiro et al. (2009)) is a functional, $\rho : \mathcal{X} \times \mathcal{T} \to \mathbb{R}$, that assigns to each random variable, $X \in \mathcal{X}$, a real value representing the degree of risk associated with $X$ with respect to some *risk attitude* or *risk tolerance*, $\tau \in \mathcal{T}$, where $\mathcal{T}$ denotes a set that captures the agent's risk attitude. In other words, a risk measure is a mapping that quantifies the risk associated with a random variable based on a given risk attitude. The precise interpretation of what 'risk' means depends on the risk measure used; different risk measures capture different aspects of a random variable. We note that it is commonplace to write $\rho(X)$ or $\rho_\tau(X)$ as shorthand for $\rho(X, \tau)$.

In essence, one can think of a risk measure as any functional that captures distributional characteristics of a random variable, typically beyond just its mean. However, an emphasis is usually placed on deriving risk measures that satisfy certain mathematical properties that can be meaningful in risk-based decision-making contexts. In Appendix A, we provide formal definitions of the various classes of risk measures used in the context of RL, where each (non-mutually-exclusive) class of risk measures can be thought of as satisfying a specific set of mathematical properties.

#### 3.2.1 NON-CONTINUAL RISK-AWARE REINFORCEMENT LEARNING

Traditionally, non-continual risk-aware RL works have aimed to optimize either a *static* risk measure (e.g. Mead et al. (2025)), or a *nested* risk measure (e.g. Wang and Delage (2025)). We now provide

informal definitions of these two classes of risk measures, as well as an informal definition pertaining to the notion *time consistency* as it relates to risk measures (see Appendix A for formal definitions):

**Static Risk Measures (Definition A.2)**: A *static* risk measure, $\rho_N$, evaluates risk at a fixed point in time, $N$. In RL-based contexts, static risk measures can be useful for quantifying the risk of the return at the end of an episode. The primary appeal of static risk measures is their interpretability.

**Nested Risk Measures (Definition A.6)**: A *nested* risk measure is a type of *dynamic* risk measure (i.e., a sequence of one-step risk measures, $\{\rho_t\}_{t=0}^N$; see Definition A.4), that is constructed recursively from the one-step risk measures, such that the risk at time $n$, $\rho_n$, can be computed as $\rho_n(X) \doteq \rho_n\big(\rho_{n+1}(\cdots \rho_N(X)\cdots)\big)$. Nested risk measures are useful because they ensure *time consistency* (described below). In RL-based contexts, nested risk measures are also useful because they can induce Bellman-like recursions with appealing dynamic programming-like properties. However, nested risk measures are typically hard to interpret. We note that typically in the RL literature, the terms 'dynamic risk measure' and 'nested risk measure' are used interchangeably; however, formally speaking, dynamic risk measures need not be time-consistent, nor have the nested structure.

**Time-Consistent Risk Measures (Definition A.5)**: A dynamic risk measure, $\{\rho_t(X)\}_{t=0}^N$, is said to be *time-consistent* if, for all $X, X' \in \mathcal{X}$ and all $t < N$, we have that $\rho_{t+1}(X) \leq \rho_{t+1}(X') \implies \rho_t(X) \leq \rho_t(X')$. Time-consistent risk measures can be appealing because they ensure that if one future outcome is deemed less risky than another at some time step, $t$, then that same outcome is not deemed more risky than the other at any other time step. One way to think about time consistency in RL-based contexts is to ask the question: *can the agent change its mind about how risky something is based on new information?* If the answer is *yes*, then there exists a lack of time consistency.

## 4 Continual Risk-Aware Reinforcement Learning

What does it mean for an RL agent to be risk-aware in a continual setting? In this section, we seek to answer this question by conducting the first formal theoretical treatment of risk-awareness in a continual setting. In particular, in Section 4.1, we formalize the notion of risk-awareness in a continual setting, and propose a set of axioms that establish the conditions needed to enable an agent to be risk-aware in a continual setting. Then in Section 4.2, we show that both of the existing classes of risk measures used in non-continual risk-aware RL (i.e., static and nested) are incompatible with these axioms. Finally, in Section 4.3, we propose a new class of *ergodic* risk measures, along with a corresponding RL objective, and show that both are compatible with continual learning.

### 4.1 Risk-Awareness and Continual Reinforcement Learning

In this section, we formalize the notion of risk-awareness in a continual setting. More specifically, we propose a risk-aware generalization of the continual RL framework proposed in Abel et al. (2025), such that we formalize risk-aware continual RL as an ongoing exchange of observations and actions between an agent and an environment, such that the agent receives observations from the environment, processes those observations into *risk-aware observations* based on its *risk attitude* (or *risk tolerance*), then emits actions back to the environment based on the risk-aware observations. We then leverage this risk-aware framework to propose a set of axioms that establish the conditions needed to enable an agent to be risk-aware in a continual setting.

To this end, let us begin by considering the aforementioned framework for continual RL proposed in Abel et al. (2025) and outlined in Section 3.1 of this work. Under this framework, one possible interpretation of the agent's *goal* (or *objective*) is as follows:

**Proposition 4.1.** *Under the continual RL framework proposed in Abel et al. (2025), the agent's goal (or objective) can be interpreted as wanting to leverage past observations from the environment, $O_{a:b}$, to find and output a sequence of actions, $A_{c:d}$, that will result in a future sequence of observations from the environment that optimizes some scalar objective, $J$.*

Less formally, the above proposition states that the agent's goal is to optimize some scalar objective, $J$. From a mathematical perspective, this $J$ is some functional, $J : \mathcal{O} \to \mathbb{R}$, that assigns to each observation, $O_t \in \mathcal{O}$, a scalar value, such that the agent's goal is to induce a sequence of observations from the environment that optimizes this scalar value.

Now, let us consider this objective in the context of a *risk-aware* agent. In particular, when one examines the RL literature as it relates to risk-aware decision-making (e.g. see Section 2.2), there are two universal functions that a risk-aware agent needs to perform. First, the agent needs to be able to *gauge* or *assess* the risk of a given sequence of observations from the environment with respect to a given *risk attitude* or *risk tolerance*. Then, based on this risk assessment, the agent needs to find and output a sequence of actions that will result in a future sequence of observations whose risk assessment is optimal with respect to the agent's risk attitude. Importantly, if we take the output of the agent's risk assessment to be a scalar value, then the risk assessment process itself can be viewed as the mapping $\mathcal{O} \times \mathcal{T} \to \mathbb{R}$, where $\mathcal{T}$ denotes a set that captures the agent's risk attitude. Accordingly, if we interpret $\mathcal{O} \times \mathcal{T}$ as a *risk-aware observation*, then the process of optimizing this observation (i.e., the assessed risk) can be viewed as a risk-aware generalization of the optimization process described in Proposition 4.1, where, by definition, $J$ corresponds to a risk measure, $\rho$ (see Section 2.2). This is formalized as Proposition 4.2 below:

**Proposition 4.2.** *If the output of the agent's risk assessment is a scalar, then this scalar corresponds to a risk measure, and the act of optimizing this risk measure can be viewed as a risk-aware generalization of the optimization process described in Proposition 4.1, such that the agent receives observations from the environment, processes those observations into risk-aware observations based on its risk attitude, then emits actions back to the environment based on the risk-aware observations.*

As such, we have essentially derived a risk-aware generalization of the continual RL framework proposed in Abel et al. (2025) (see Figure 1b)). Accordingly, we can also generalize the definition of plasticity (1) for a risk-aware agent as follows:

$$\mathfrak{R}_{\substack{a:b \\ c:d}} \doteq \mathbb{I}(O_{a:b} \times \tau_{a:b} \to A_{c:d}), \tag{2}$$

where $O_{a:b} \times \tau_{a:b}$ denotes a sequence of *risk-aware* observations with respect to the agent's risk attitude, $\tau_t \in \mathcal{T}$, and $\mathfrak{R}$ denotes the *risk-aware plasticity*.

Before proceeding, we want to emphasize the point made about $\rho$ being a generalization of $J$. In particular, if we take $\rho(X, \tau) = \mathbb{E}[X]$ (either by explicitly setting $\rho = \mathbb{E}$, or based on a specific choice of $\tau \in \mathcal{T}$ that results in $\rho(X, \tau) = \mathbb{E}[X]$), then we recover the risk-neutral objective, $J$. Accordingly, we note that any result or discussion point that is made with reference to a risk-aware objective, $\rho$, in this work is also applicable to any risk-neutral objective, $J$.

### 4.1.1 Axioms for Risk-Awareness in the Continual Setting

Having formalized the notion of risk-awareness in the context of continual RL, we now seek to establish the minimum requirements needed for the agent to be able to perform the aforementioned risk assessment and optimization in a continual learning setting. To this end, we argue that any limitations related to the risk assessment and subsequent optimization should be direct consequences of the limitations of the agent itself. In particular, we focus on two well-argued limitations (or requirements) of continual RL agents (e.g. see Abel et al. (2023), Kumar et al. (2025)), which are formalized as Assumption 4.1 below:

**Assumption 4.1.** *Any continual RL agent has a finite memory, and has a non-zero level of plasticity.*

As such, based on the above limitations of the agent, we conclude this section by proposing the following axioms as the minimum requirements needed for risk-awareness in the continual setting:

**Proposition 4.3.** *Under Assumption 4.1, the following two axioms are necessary and sufficient conditions for risk-awareness in the continual setting:*

1. *Feasibility Axiom: The agent's risk assessment must be computable based on the agent's finite memory. That is, the risk assessment must be computable based on observations collected over a finite time interval $[a : b]$ with respect to the agent's risk attitude during that same time interval.*

2. *Plasticity Axiom: $\mathbb{P}(\mathfrak{R} > 0$ infinitely often$) = 1$. In words, this axiom states that the agent's risk assessment and subsequent behaviour should be influenced by changes in its risk attitude as well as new observations from the environment infinitely often.*

## 4.2 Existing Risk Measures and Continual Reinforcement Learning

In the previous section, we formalized the notion of risk-awareness in the continual setting, and showed that risk measures are a natural fit as an objective in this setting. But how exactly should we go about designing a risk measure? That is, how can we design a risk measure that satisfies the axioms in Proposition 4.3, and is also a sensible and 'good' measure of risk?

A natural first step towards answering this question could be to examine the existing classes of risk measures used in non-continual risk-aware RL (see Section 3.2.1) and see if we can leverage them in the continual setting. To this end, upon examining the existing classes of risk measures used in non-continual risk-aware RL (i.e., static and nested), we find that they are incompatible with the axioms established in Proposition 4.3. This is formally shown below:

**Lemma 4.1.** *Let $\rho_N$ denote a static risk measure (as per Definition A.2) defined for some time step, $N$. The static risk measure, $\rho_N$, does not satisfy both axioms of Proposition 4.3.*

*Proof.* As $N \to \infty$, this violates the Feasibility axiom as the computation of the risk would require the agent to have infinite memory. If $N < \infty$, then any observation that occurs after time step $N$ would not influence the risk assessment, thereby violating the Plasticity axiom. $\qquad\square$

**Lemma 4.2.** *Let $\{\rho_t\}_{t=0}^{\infty}$ denote a nested risk measure and let $\rho_n$ denote the assessed risk at time step $n$, such that its value can be computed as: $\rho_n(X) \doteq \rho_n\big(\rho_{n+1}(\cdots)\big)$ (see Definition A.6). The nested risk measure, $\{\rho_t\}_{t=0}^{\infty}$, does not satisfy both axioms of Proposition 4.3.*

*Proof.* Consider the risk assessment at time $n$, $\rho_n(X) \doteq \rho_n\big(\rho_{n+1}(\cdots)\big)$. By definition, the validity of this risk calculation is contingent on time consistency holding for all $t \geq n$. However, this implies that future observations cannot influence how risky $X$ is relative to other outcomes. Hence, either time consistency is satisfied, which violates the Plasticity axiom, or time consistency is violated, which invalidates the risk calculation at time $n$, thereby violating the Feasibility axiom. $\qquad\square$

## 4.3 Ergodic Risk Measures for Continual Reinforcement Learning

In the previous section, we showed that static and nested risk measures are not compatible with continual learning. Based on this insight, we now propose a new class of *ergodic* risk measures, along with a corresponding RL objective, and show that they are both compatible with continual learning. To this end, we begin by formally defining our proposed class of ergodic risk measures:

---

**Definition 4.1.** *(Ergodic Risk Measure) Let $\{\rho_t\}_{t=0}^{\infty}$ denote a sequence of conditional risk measures (i.e., a dynamic risk measure) defined on a probability space $(\Omega, \mathcal{F}, \mathbb{P})$ with filtration $(\mathcal{F}_t)_{t=0}^{\infty}$, where $\mathcal{F}_t \subseteq \mathcal{F}$ denotes the $\sigma$-algebra representing the information collected up until time step $t \geq 0$, such that $\rho_t : L^{\infty}(\mathcal{F}_{\infty}) \to L^{\infty}(\mathcal{F}_t)$, where $L^{\infty}(\mathcal{F}_u)$ denotes the space of essentially bounded, $\mathcal{F}_u$-measurable random variables.*

*Suppose that for any time step, $t > 0$, there exists a finite time interval, $[n : t]$, such that for all $X \in L^{\infty}(\mathcal{F}_{\infty})$, and some $\hat{\rho}_t(X) \in L^{\infty}(\mathcal{G}_{n:t})$, where $\mathcal{G}_{n:t}$ denotes the $\sigma$-algebra generated by the information from time step $n$ up to time step $t$, we have that:*

$$\rho_t(X) - \hat{\rho}_t(X) \approx 0. \qquad (3)$$

*That is, the risk assessment at any given time step, $t$, can be accurately computed based on a preceding, finite subset of the history, such that the risk assessment effectively ceases to depend on the history that occurs prior to the interval $[n : t]$. Furthermore, suppose that $\{\rho_t\}_{t=0}^{\infty}$ is designed in such a way that there exists some time step, $t$, such that time consistency (Definition A.5) does not hold. Then, we call $\{\rho_t\}_{t=0}^{\infty}$ an* ergodic *risk measure.*

---

In essence, an ergodic risk measure is a (non-nested) dynamic risk measure, such that its one-step risk assessments, $\rho_t$, can be *accurately* computed using only observations collected over a rolling finite time interval, $[n : t]$. We note that the size of this interval can vary between different time steps. Figure 2 depicts a comparison between static, nested, and ergodic risk measures. In particular, the figure visualizes which observations are needed to accurately compute the risk at a given time step.

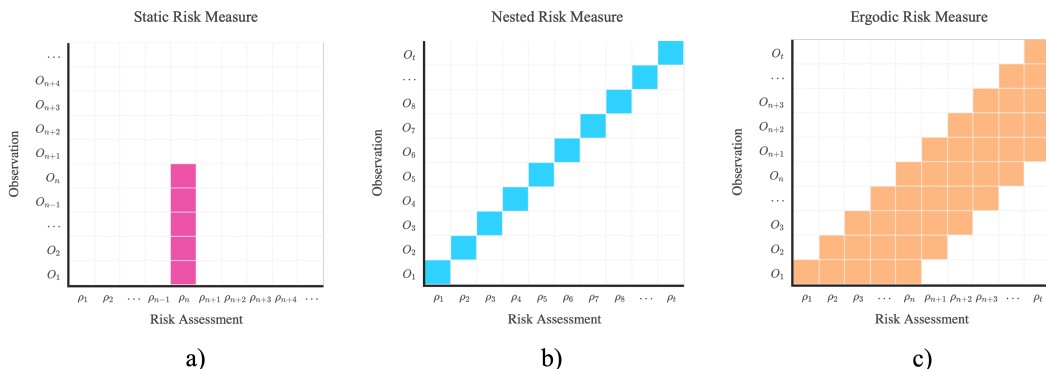

Figure 2: A comparison between: **a)** static, **b)** nested, and **c)** ergodic risk measures in terms of which observations are needed to accurately compute the risk at a given time step. A shaded box indicates that the observation (as per the y-axis value) is needed to accurately compute the risk at that time step (x-axis value). Note that the nested risk measure depicted in this figure is assumed to be a *Markov* risk measure (see Definition A.7).

We now show that ergodic risk measures satisfy both of the axioms from Proposition 4.3:

**Lemma 4.3.** *Let $\{\rho_t\}_{t=0}^\infty$ denote an ergodic risk measure as defined in Definition 4.1. The ergodic risk measure, $\{\rho_t\}_{t=0}^\infty$, satisfies both axioms of Proposition 4.3.*

*Proof.* Since an ergodic risk measure is a dynamic risk measure such that time consistency (as defined in Definition A.5) does not hold, it satisfies the Plasticity axiom since the risk assessments and subsequent behaviour will be influenced by new observations and any changes in the agent's risk attitude infinitely often. Similarly, by definition, the risk at any time $t$ can be computed using observations collected over a finite time interval with respect to the agent's risk attitude during that same time interval, thereby satisfying the Feasibility axiom. This completes the proof. □

### 4.3.1 AN ERGODIC REINFORCEMENT LEARNING OBJECTIVE

Having established that ergodic risk measures are compatible with continual learning, we now motivate a corresponding RL objective that can be used to perform risk-aware learning and decision-making in the continual setting. To this end, we will utilize the average-reward MDP framework (Puterman, 1994) to realize the agent-environment interaction described in Section 3.1. Our choice of the average-reward MDP formulation in this work follows prior work, such as Sharma et al. (2022) and Kumar et al. (2025), which argue that the average-reward formulation's emphasis on long-term performance is a natural fit for continual learning settings.

More formally, consider a finite MDP, $\mathcal{M} \doteq \langle \mathcal{S}, \mathcal{A}, \mathcal{R}, p \rangle$, where $\mathcal{S}$ is a finite set of states, $\mathcal{A}$ is a finite set of actions, $\mathcal{R} \subset \mathbb{R}$ is a bounded set of rewards, and $p : \mathcal{S} \times \mathcal{A} \times \mathcal{R} \times \mathcal{S} \to [0, 1]$ is a probabilistic transition function that describes the dynamics of the environment, such that at each discrete time step, $t = 0, 1, 2, \ldots$, an agent chooses an action, $A_t \in \mathcal{A}$, based on its current state, $S_t \in \mathcal{S}$, and receives a reward, $R_{t+1} \in \mathcal{R}$, while transitioning to a (potentially) new state, $S_{t+1}$, such that $p(s', r \mid s, a) = \mathbb{P}(S_{t+1} = s', R_{t+1} = r \mid S_t = s, A_t = a)$.

Under this framework, the continual RL problem can be viewed as an infinite sequence of MDPs, $\{\mathcal{M}^k\}_{k=1}^\infty$, such that $\mathcal{M}^k \doteq \langle \mathcal{S}_k, \mathcal{A}_k, \mathcal{R}_k, p_k \rangle$, where each $\mathcal{M}^k$ may differ in its state-space, action-space, reward function, and/or transition dynamics based on some indexing function, $\omega : \mathbb{N} \to \mathbb{N}$, such that $\omega(t) = i$ indicates that at time step $t$, the agent is in MDP $\mathcal{M}^i$.

The agent's goal in this framework is to construct a sequence of stationary policies, $\{\pi^k\}_{k=1}^\infty$, that optimizes the long-run (or limiting) average-reward, $\bar{r}$, which is defined as follows for a given stationary policy being followed at time $t$, $\pi_t \doteq \pi^{\omega(t)}$:

$$\bar{r}_{\pi_t}(s) \doteq \lim_{h\to\infty} \frac{1}{h} \sum_{n=n_0}^{h} \mathbb{E}[R_n \mid S_{n_0} = s, A_{n_0:n-1} \sim \pi^{\omega(n_0)}], \tag{4}$$

where $n_0 = \min\{y : \pi^{\omega(y)} = \pi_t\}$. When working with average-reward MDPs, it is common to simplify the expression for the average-reward objective (4) into a more workable form by making certain *ergodicity-like* assumptions about the Markov chain induced when following policy $\pi_t$. To this end, a *unichain* assumption is typically used when doing prediction (learning) because it ensures the existence of a unique limiting distribution of states, $\mu_{\pi_t}(s) \doteq \lim_{n\to\infty} \mathbb{P}(S_n = s \mid A_{n_0:n-1} \sim \pi^{\omega(n_0)})$, that is independent of initial conditions. Similarly, a *communicating* assumption is typically used for control (optimization) because it ensures the existence of a unique optimal average-reward, $\bar{r}*$, that is independent of initial conditions. Importantly, these ergodicity-like assumptions enable the agent's objective to be expressed as a stable measure of long-term performance that eventually becomes independent of prior conditions (i.e., $\bar{r}_{\pi_t}(s) = \bar{r}_{\pi_t}$).

Hence, to perform risk-aware learning and decision-making in a continual setting, we can consider the risk-aware analogue to the (risk-neutral) average-reward RL objective (4):

$$\rho_{\pi_t}(s) \doteq \lim_{h\to\infty} \frac{1}{h} \sum_{n=n_0}^{h} \rho_\tau[R_n \mid S_{n_0} = s, A_{n_0:n-1} \sim \pi^{\omega(n_0)}]. \tag{5}$$

That is, we want to optimize some risk measure, $\rho$, pertaining to the limiting per-step reward distribution induced when following a given (stationary) policy, $\pi_t$. However, the risk measure presented in Equation (5) is dependent on the initial conditions, which is not tractable in the continual setting. As such, as with the average-reward objective (4), we can apply an appropriate ergodicity-like assumption that makes the risk-aware objective independent of prior conditions (i.e., $\rho_{\pi_t}(s) = \rho_{\pi_t}$):

**Assumption 4.2** (Unichain Assumption for Prediction). *For a given MDP, $\mathcal{M}^k$, the Markov chain induced by the policy, $\pi^k$, is unichain. That is, the induced Markov chain consists of a single recurrent class and a potentially-empty set of transient states.*

**Assumption 4.3** (Communicating Assumption for Control). *For a given MDP, $\mathcal{M}^k$, the MDP has a single communicating class. That is, each state in the MDP is accessible from every other state in the MDP under some deterministic stationary policy.*

For clarity, we note that the above assumptions need only apply independently for each MDP, $\mathcal{M}^k$, rather than holistically for the entire sequence of MDPs, $\{\mathcal{M}^k\}_{k=1}^{\infty}$. For example, Assumption 4.3 only requires that each state in a given MDP, $\mathcal{M}^k$, is accessible from every other state in $\mathcal{M}^k$, but not the states of other MDPs, $\mathcal{M}^{i \neq k}$.

We now show that, under the above ergodicity-like assumptions, the risk-aware objective (5) corresponds to an ergodic risk measure, thereby making it compatible with continual learning:

> **Theorem 4.1.** *Given an appropriate ergodicity-like assumption, such as Assumption 4.2 or 4.3, and a stationary policy, $\pi_t$, the risk-aware objective (5) corresponds to an ergodic risk measure, as defined in Definition 4.1.*

*Proof.* We begin by noting that the risk-aware objective (5) clearly assesses risk over the entire time horizon and does not enforce time consistency (as per Definition A.5), thereby satisfying the definition of a dynamic, non time-consistent risk measure.

Next, we show that for any time step, $t$, we can accurately compute the risk based on observations collected over a finite time interval. To this end, consider an MDP, $\mathcal{M}^k$, through which an agent interacts with the environment for the possibly-infinite time interval $[a : b]$ (note that $a$ here is equivalent to $n_0$ in Equation (5)). If $[a : b]$ is finite, then the desired condition is automatically satisfied. If $[a : b]$ is infinite, then, under the ergodicity-like assumption, we have, by Birkhoff's Ergodic Theorem (Birkhoff, 1931), that $\rho_{\pi_t}$ converges to a stationary value as $t \to \infty$. This implies that there exists a finite time step, $j$, such that for all $t \geq j$, $\rho_{\pi_t}$ converges to a stationary value. We therefore have two cases to check: 1) If $t \leq j$, then, since $j$ is finite, the risk can be computed based on observations collected over the finite time interval $[a : j]$, thereby satisfying the desired condition. 2) If $t > j$, then the convergence of $\rho_{\pi_t}$ to a stationary value as $t \to \infty$ implies that for any $t > j$, we can find a finite time interval, $[n : t]$ (where $n \geq j$), that can be used to accurately compute the risk assessment, thereby satisfying the desired condition.

As such, the definition of an ergodic risk measure (4.1) is satisfied. This completes the proof. $\square$

We conclude this section by providing some remarks related to ergodic risk measures:

**Remark 4.1.** *Ergodic risk measures are also compatible with the generic, possibly non-continual average-reward setting. That is, ergodic risk measures provide a formalism for risk-aware objectives of the form (5), which had been studied previously in non-continual average-reward settings (e.g. Xia et al. (2023), Rojas and Lee (2025)), but never formalized in the context of risk measure theory.*

**Remark 4.2.** *As per Definition 4.1, time consistency (as defined in Definition A.5) does not hold for ergodic risk measures. However, we note that it can be easily checked that, under ergodicity-like assumptions, we can recover a weaker notion of time consistency for the risk-aware objective (5) as $t \to \infty$ (that is, the time consistency condition (A.1) holds as $t \to \infty$, but not for all $t$, as is required by Definition A.5). Accordingly, this shows that it is still possible to recover some (weaker) notion of time consistency with ergodic risk measures.*

## 5    CASE STUDY: CVAR AS AN ERGODIC RISK MEASURE

In this section, we present a case study in which we show how optimizing an ergodic risk measure can induce sensible risk-aware behaviour in a continual learning setting. That is, we show empirically how optimizing an ergodic risk measure can enable the agent to successfully adapt in situations where either its risk attitude changes, or the observations from the environment change.

To this end, we focus on optimizing the well-known conditional value-at-risk (CVaR) risk measure (Rockafellar and Uryasev, 2000). More formally, consider a random variable $X$ with a cumulative distribution function, $F(x) = \mathbb{P}(X \le x)$. The (left-tail) *value-at-risk (VaR)* of $X$ with parameter $\tau \in (0, 1)$ represents the $\tau$-quantile of $X$, such that $\text{VaR}_\tau(X) = \sup\{x \mid F(x) \le \tau\}$. When $F(x)$ is continuous at $x = \text{VaR}_\tau(X)$, $\text{CVaR}_\tau(X)$ can be interpreted as the expected value of $X$ conditioned on $X \le \text{VaR}_\tau(X)$, such that $\text{CVaR}_\tau(X) = \mathbb{E}[X \mid X \le \text{VaR}_\tau(X)]$. Importantly, as per the results in Section 4, and given Assumptions 4.2 and 4.3, the CVaR risk measure can be formulated as the following continual learning objective that corresponds to an ergodic risk measure:

$$\text{CVaR}_{\pi_t} \doteq \lim_{h \to \infty} \frac{1}{h} \sum_{n=n_0}^{h} \text{CVaR}_\tau[R_n \mid S_n \sim \mu_{\pi^{\omega(n_0)}}, A_n \sim \pi^{\omega(n_0)}]. \tag{6}$$

In this case study, we optimize the CVaR objective (6) via the *RED CVaR Q-learning* algorithm proposed in Rojas and Lee (2025) (see Remark 4.1) in two continual learning tasks.

In the first task, we consider a continual variation of the *red-pill blue-pill (RPBP)* task (Rojas and Lee, 2025). More specifically, in the regular (non-continual) RPBP task, an agent, at each time step, can take either a 'red pill', which takes them to the 'red world' state, or a 'blue pill', which takes them to the 'blue world' state. Each state has its own characteristic per-step reward distribution, such that for a sufficiently low CVaR parameter, $\tau$, the red world state has a reward distribution with a lower (worse) mean but a higher (better) CVaR compared to the blue world state. In the continual variation of RPBP considered in this task, the risk attitude of the agent, which is governed by the CVaR parameter, $\tau$, changes over time from risk-neutral ($\tau \approx 1$) to risk-averse ($\tau \approx 0$). In particular, we would expect that the agent first learns to stay in the blue world state, but then changes its preference to the red world state as its risk attitude changes from risk-neutral to risk-averse. More formally, this task can be viewed as a continual learning task with a changing reward function (see Appendix B for more details). We refer to this task as the $\tau$-RPBP task.

In the second task, we consider another continual variation of the RPBP task. In this variation, the characteristic per-step reward distributions of the states change over time, such that the agent is required to continually adapt and find the state with the better CVaR (given a fixed risk attitude, $\tau$). More formally, this task can be viewed as a continual learning task with a changing state-space (such that a given state is effectively replaced with a state with a different per-step reward distribution; see Appendix B for more details). We refer to this task as the $\mathcal{S}$-RPBP task.

In terms of empirical results, Figures 3 and 4 show the resulting agent behaviour as learning progresses in both tasks. In particular, Figure 3 shows that in the $\tau$-RPBP task, the agent correctly learns to stay in the blue world state in the beginning, and then correctly changes its preference to the red world state once its risk attitude changes from risk-neutral to risk-averse. Similarly, Figure 4 shows that in the $\mathcal{S}$-RPBP task, the agent is able to continually adapt and find the state with the better CVaR. The full set of experimental details and results can be found in Appendix B.

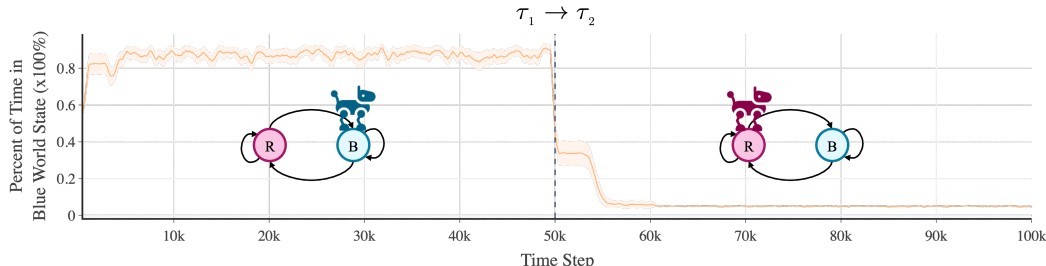

Figure 3: Rolling percent of time that the agent stays in the blue world state as learning progresses in the $\tau$-RPBP task. A solid line denotes the mean percent of time spent in the blue world state, and the shaded region denotes a 95% confidence interval over 50 runs. As shown in the figure, the agent correctly learns to stay in the blue world state in the beginning, and then correctly changes its preference to the red world state once its risk attitude changes from risk-neutral to risk-averse.

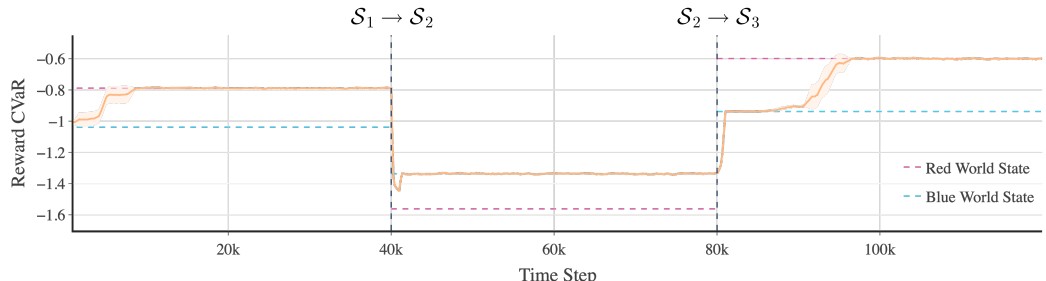

Figure 4: Rolling reward CVaR as learning progresses in the $\mathcal{S}$-RPBP task. A solid line denotes the mean CVaR, and the shaded region denotes a 95% confidence interval over 10 runs. The blue and red dashed lines denote the reward CVaR of the blue and red world states, respectively. As shown in the figure, the agent is able to continually adapt and find the state with the better CVaR.

## 6 DISCUSSION

In this work, we took the first steps towards developing a risk-aware foundation for continual RL. In particular, we formalized the notion of risk-awareness in a continual setting, and used the resulting framework to propose a set of axioms that establish the conditions needed to enable an agent to be risk-aware in a continual setting. We then examined the classical theory of risk measures, and showed that, in its current form, it is incompatible with continual learning. Then, building on this insight, we extended risk measure theory into the continual setting by introducing a new class of *ergodic* risk measures that are compatible with continual learning. Finally, we provided a CVaR-based case study, along with numerical results, which showed the intuitive appeal of ergodic risk measures in continual settings.

More broadly, the introduction of ergodic risk measures offers a new alternative for risk-aware learning and decision-making, even in non-continual settings (such as in the non-continual average-reward setting). Moreover, in comparison to the static and nested (dynamic) risk measures that are typically used in non-continual RL settings, we note that ergodic risk measures offer several advantages. In particular, ergodic risk measures retain some notion of time consistency, while remaining highly interpretable, thereby capturing the appeal of both static and nested risk measures.

All in all, this work represents the first formal theoretical exploration of risk-aware decision-making in a continual learning setting. Moving forward, we believe that the theoretical foundation that has been established, including the introduction of a theoretically-sound risk-aware objective that is stable-yet-adaptable, will enable further progress in the development of risk-aware lifelong agents.

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

## A    RISK MEASURES

In this appendix, we provide formal definitions of the various classes of risk measures used in the context of RL, where each (non-mutually-exclusive) class of risk measures can be thought of as satisfying a specific set of mathematical properties:

**Definition A.1.** *(Coherent Risk Measure; adapted from Artzner et al. (1999)) A risk measure, $\rho$, is called coherent if it satisfies the following four axioms for all $X, X' \in \mathcal{X}$:*

1. *Monotonicity: If $X \leq X'$ almost surely, then $\rho(X) \leq \rho(X')$.*

2. *Translation Invariance: For all $c \in \mathbb{R}$, $\rho(X + c) = \rho(X) + c$.*

3. *Positive Homogeneity: For all $\lambda \geq 0$, $\rho(\lambda X) = \lambda \rho(X)$.*

4. *Subadditivity: $\rho(X + X') \leq \rho(X) + \rho(X')$.*

Coherent risk measures are useful because they enforce a form of self-consistency in how risk is quantified and compared. In particular, monotonicity ensures that if a random variable, $X$, always yields outcomes that are no worse than outcomes induced by another random variable, $X'$, then $X$ should be considered less risky than $X'$. Translation invariance requires that adding a constant amount to $X$ simply shifts its risk by that same amount. Positive homogeneity enforces scale-consistency, such that doubling the size of $X$ also doubles its risk. Finally, subadditivity formalizes the idea of diversification, requiring that the risk of two combined random variables cannot exceed the sum of their individual risks. We note that the positive homogeneity and subadditivity properties also ensure that coherent risk measures are convex.

**Definition A.2.** *(Static Risk Measure; adapted from Artzner et al. (1999)) Let $(\Omega, \mathcal{F}, \mathbb{P})$ denote a probability space, and let $\mathcal{F}_N \subseteq \mathcal{F}$ denote the $\sigma$-algebra representing information available at time $N$. Denote by $L^\infty(\mathcal{F}_N)$ the space of essentially bounded, $\mathcal{F}_N$-measurable random variables. A static risk measure is a mapping, $\rho : L^\infty(\mathcal{F}_N) \to \mathbb{R}$, that assigns to each random variable, $X \in L^\infty(\mathcal{F}_N)$, a single real value.*

In essence, a static risk measure evaluates risk at a fixed point in time, without taking into consideration the temporal evolution of information. In RL-based contexts, static risk measures can be useful for quantifying the risk associated with the return at the end of an episode. One of the primary appeals of static risk measures is that they are considered to be easily interpretable.

**Definition A.3.** *(Conditional Risk Measure; adapted from Ruszczyński (2010)) Let $(\Omega, \mathcal{F}, \mathbb{P})$ denote a probability space with filtration $(\mathcal{F}_t)_{t=0}^N$, where $\mathcal{F}_t \subseteq \mathcal{F}$ represents the information available up to time $t$. Denote by $L^\infty(\mathcal{F}_N)$ the space of essentially bounded, $\mathcal{F}_N$-measurable random variables, and by $L^\infty(\mathcal{F}_t)$ the space of essentially bounded, $\mathcal{F}_t$-measurable random variables. A conditional risk measure at time $t$ is a mapping, $\rho_t : L^\infty(\mathcal{F}_N) \to L^\infty(\mathcal{F}_t)$, that assigns to each random variable, $X \in L^\infty(\mathcal{F}_N)$, a conditional risk evaluation, $\rho_t(X)$, that is $\mathcal{F}_t$-measurable and satisfies the following monotonicity property: if $X \leq X'$ almost surely, then $\rho_t(X) \leq \rho_t(X')$.*

In essence, a conditional risk measure at time $t$ is a mapping that evaluates the risk of future outcomes (e.g. at time $N > t$) based on the information available up to and including time $t$. The monotonicity property ensures that if a future outcome, $X$, always yields less loss (or more reward) than another outcome, $X'$, then $X$ is never assigned a higher risk than $X'$.

**Definition A.4.** *(Dynamic Risk Measure; adapted from Ruszczyński (2010)) Let $(\Omega, \mathcal{F}, \mathbb{P})$ denote a probability space with filtration $(\mathcal{F}_t)_{t=0}^N$. A dynamic risk measure is a sequence of conditional risk measures, $\{\rho_t\}_{t=0}^N$, where each $\rho_t : L^\infty(\mathcal{F}_N) \to L^\infty(\mathcal{F}_t)$ assigns to every random variable $X \in L^\infty(\mathcal{F}_N)$ a conditional risk evaluation, $\rho_t(X)$, that is $\mathcal{F}_t$-measurable.*

In essence, a dynamic risk measure provides a time-indexed family of risk assessments, which allows risk to be tracked and updated as new information becomes available. In RL-based contexts, dynamic risk measures can be useful for capturing the sequential nature of decision-making (i.e., that actions taken at each time step can potentially influence future outcomes, and hence, future risk evaluations).

**Definition A.5.** *(Time-Consistent Risk Measure; adapted from Boda and Filar (2006)) Let $\{\rho_t\}_{t=0}^N$ be a dynamic risk measure defined on a probability space $(\Omega, \mathcal{F}, \mathbb{P})$ with filtration $(\mathcal{F}_t)_{t=0}^N$. The dynamic risk measure is said to be time-consistent if, for all random variables $X, X' \in L^\infty(\mathcal{F}_N)$ and all $t < N$,*

$$\rho_{t+1}(X) \leq \rho_{t+1}(X') \implies \rho_t(X) \leq \rho_t(X'). \tag{A.1}$$

Time-consistent risk measures can be appealing because they ensure that if one future outcome is deemed less risky than another at some time step, $t$, then that same outcome is not deemed more risky than the other at any other time step. One way to think about time consistency in RL-based contexts is to ask the question: *can the agent change its mind about how risky something is based on new information?* If the answer is *yes*, then there exists a lack of time consistency.

**Definition A.6.** *(Nested Risk Measure; adapted from Ruszczyński (2010)) A nested risk measure is a dynamic risk measure, $\{\rho_t\}_{t=0}^N$, that is constructed recursively from one-step conditional risk measures. More formally, given conditional risk measures, $\rho_t : L^\infty(\mathcal{F}_{t+1}) \to L^\infty(\mathcal{F}_t)$, the risk at time n, $\rho_n$, can be evaluated as:*

$$\rho_n(X) \doteq \rho_n\big(\rho_{n+1}(\cdots \rho_N(X) \cdots)\big). \tag{A.2}$$

Nested risk measures are useful because they ensure time consistency. That is, the risk evaluation at earlier times is consistent with future evaluations. In RL-based contexts, nested risk measures are also useful because they can induce Bellman-like recursions with appealing dynamic programming-like properties. One of the drawbacks of nested risk measures is that they are typically hard to interpret. We note that typically in the RL literature, the terms 'dynamic risk measure' and 'nested risk measure' are used interchangeably; however, formally speaking, dynamic risk measures need not be time-consistent, nor have the nested structure.

**Definition A.7.** *(Markov Risk Measure; adapted from Ruszczyński (2010)) Let $(\Omega, \mathcal{F}, \mathbb{P})$ denote a probability space, and let $\{S_t\}_{t=0}^N$ denote a Markov process where each state, $S_t : \Omega \to \mathcal{S}$, takes values in a measurable state-space $(\mathcal{S}, \mathcal{B}(\mathcal{S}))$, such that $\mathcal{F}_t \doteq \sigma(S_0, \ldots, S_t)$. Here, $\mathcal{B}(\mathcal{S})$ denotes the Borel $\sigma$-algebra on $\mathcal{S}$. A one-step conditional risk measure, $\rho_t : L^\infty(\mathcal{F}_{t+1}) \to L^\infty(\mathcal{F}_t)$, is called a Markov risk measure if, for every $X \in L^\infty(\mathcal{F}_{t+1})$, the risk assessment satisfies $\rho_t(X) \in L^\infty(\sigma(S_t))$, where $\sigma(S_t) \subseteq \sigma(S_0, \ldots, S_t) = \mathcal{F}_t$. That is, the risk assessment $\rho_t(X)$ is $\sigma(S_t)$-measurable, such that it only depends on the current state, $S_t$.*

Markov risk measures are useful because they enforce a one-step time dependence structure that makes them compatible with MDP-based RL solution methods. From a risk perspective, this means that the assessment of risk at each time step only depends on the information available at that time step, rather than the entire history of past information. Note that in an MDP setting (as opposed to the simpler Markov process described in Definition A.7), the 'state' can be characterized as a state-action pair. That is, $\rho_t(X) \in L^\infty(\sigma(S_t, A_t))$ for some $A_t$ in a measurable action-space, $\mathcal{A}$.

# B  NUMERICAL EXPERIMENTS

This appendix contains details regarding the numerical experiments performed as part of this work. The overall aim of the experiments was to provide a concrete example of an ergodic risk measure being optimized in a continual learning setting. In particular, we focused on the well-known conditional value-at-risk (CVaR) risk measure (Rockafellar and Uryasev, 2000). More formally, consider a random variable $X$ with a finite mean on a probability space $(\Omega, \mathcal{F}, \mathbb{P})$, and with a cumulative distribution function $F(x) = \mathbb{P}(X \leq x)$. The (left-tail) *value-at-risk (VaR)* of $X$ with parameter $\tau \in (0, 1)$ represents the $\tau$-quantile of $X$, such that $\text{VaR}_\tau(X) = \sup\{x \mid F(x) \leq \tau\}$. When $F(x)$ is continuous at $x = \text{VaR}_\tau(X)$, $\text{CVaR}_\tau(X)$ can be interpreted as the expected value of $X$ conditioned on $X$ being less than or equal to $\text{VaR}_\tau(X)$, such that $\text{CVaR}_\tau(X) = \mathbb{E}[X \mid X \leq \text{VaR}_\tau(X)]$.

As per Section 5, the CVaR risk measure can be formulated as the continual learning objective (6), which is displayed below as Equation (B.1) for convenience:

$$\text{CVaR}_{\pi_t} \doteq \lim_{h \to \infty} \frac{1}{h} \sum_{n=n_0}^{h} \text{CVaR}_\tau[R_n \mid S_n \sim \mu_{\pi^{\omega(n_0)}}, A_n \sim \pi^{\omega(n_0)}]. \tag{B.1}$$

That is, we aimed to optimize the (left-tail) conditional value-at-risk associated with the limiting per-step reward distribution induced when following stationary policy $\pi_t$. More specifically, our aim was to optimize the CVaR objective (B.1) in two continual learning tasks via the *RED CVaR Q-learning* algorithm proposed in Rojas and Lee (2025). In particular, the RED CVaR Q-learning algorithm was designed to optimize the CVaR associated with the long-run per-step reward distribution of an average-reward MDP, which precisely corresponds to the continual learning objective (B.1). The RED CVaR Q-learning algorithm (Algorithm 1) is shown below:

---

**Algorithm 1** RED CVaR Q-Learning (Tabular) (Rojas and Lee, 2025)

---

    **Input:** the policy $\pi$ to be used (e.g., $\varepsilon$-greedy)
    **Algorithm parameters:** step size parameters $\alpha$, $\alpha_{\text{CVaR}}$, $\alpha_{\text{VaR}}$; CVaR parameter $\tau$
    Initialize $Q(s, a)\ \forall s, a$ (e.g. to zero)
    Initialize CVaR arbitrarily (e.g. to zero)
    Initialize VaR arbitrarily (e.g. to zero)
    Obtain initial $S$
    **while** still time to train **do**
        $A \leftarrow$ action given by $\pi$ for $S$
        Take action $A$, observe $R, S'$
        $\tilde{R} = \text{VaR} - \frac{1}{\tau}\max\{\text{VaR} - R, 0\}$
        $\delta = \tilde{R} - \text{CVaR} + \max_a Q(S', a) - Q(S, A)$
        $Q(S, A) = Q(S, A) + \alpha\delta$
        $\text{CVaR} = \text{CVaR} + \alpha_{\text{CVaR}}\delta$
        **if** $R \geq \text{VaR}$ **then**
            $\text{VaR} = \text{VaR} + \alpha_{\text{VaR}}(\delta + \text{CVaR} - \text{VaR})$
        **else**
            $\text{VaR} = \text{VaR} + \alpha_{\text{VaR}}\left(\left(\frac{\tau}{\tau-1}\right)\delta + \text{CVaR} - \text{VaR}\right)$
        **end if**
        $S = S'$
    **end while**
    return $Q$

---

In terms of the two continual learning tasks considered in this work, we considered two continual variations of the *red-pill blue-pill (RPBP)* task (Rojas and Lee, 2025). More specifically, in the regular (non-continual) RPBP task, an agent, at each time step, can take either a 'red pill', which takes them to the 'red world' state, or a 'blue pill', which takes them to the 'blue world' state. Each state has its own characteristic per-step reward distribution, such that for a sufficiently low CVaR parameter, $\tau$, the red world state has a reward distribution with a lower (worse) mean but a higher (better) CVaR compared to the blue world state. That is, in the regular RPBP task, for a sufficiently low CVaR parameter, $\tau$, we would expect a risk-neutral agent to learn a policy that prefers to stay in the blue world, and a risk-averse agent to learn a policy that prefers to stay in the red world.

We now discuss the two continual variations of the RPBP task considered in this work:

### B.1 $\tau$-RPBP Task

In the first task, we considered a continual variation of the RPBP task, such that the *risk attitude* of the agent, which is governed by the CVaR parameter, $\tau$, changes over time from risk-neutral ($\tau = 0.9$) to risk-averse ($\tau = 0.1$). In particular, we would expect that the agent first learns to stay in the blue world state, but then changes its preference to the red world state as its risk attitude changes from risk-neutral to risk-averse. More formally, this task can be viewed as a continual learning task, $\{\mathcal{M}^k\}_{k=1}^2$, with a changing reward function, such that $\mathcal{M}^k \doteq \langle \mathcal{S}, \mathcal{A}, \mathcal{R}_k, p \rangle$, where

$$\tilde{R}_{t,k} = \text{VaR}_t - \frac{1}{\tau_k}(\text{VaR}_t - R_t)^+ \text{ (see Algorithm 1)}, \tag{B.2}$$

with $\tau_1 = 0.9$ and $\tau_2 = 0.1$. The indexing function, $\omega$, was defined such that $\omega(t) = 1$ for $t < 50,000$, and $\omega(t) = 2$ otherwise. That is, the agent's risk attitude changes from risk-neutral to risk-averse at $t = 50,000$.

In terms of the hyperparameters used with the RED CVaR Q-learning algorithm, we used the tuned hyperparameters from Rojas and Lee (2025). That is, $\alpha = 2e\text{-}2$, $\alpha_{\text{CVaR}} \doteq \eta_{\text{CVaR}}\alpha$, where $\eta_{\text{CVaR}} = 1e\text{-}1$, and $\alpha_{\text{VaR}} \doteq \eta_{\text{VaR}}\alpha$, where $\eta_{\text{VaR}} = 1e\text{-}1$. We used an $\varepsilon$-greedy policy with a fixed epsilon of 0.1, and set all initial guesses to zero. The results for this $\tau$-*RPBP* task are shown in Figure 3.

### B.2 $\mathcal{S}$-RPBP Task

In the second task, we considered another variation of the RPBP task. In this variation, the characteristic per-step reward distributions of the states change over time, such that the agent is required to continually adapt and find the state with the better CVaR (given a fixed risk attitude, $\tau$). More formally, this task can be viewed as a continual learning task with a changing state-space, such that a given state is effectively replaced with a state with a different per-step reward distribution. That is, we have a continual learning task, $\{\mathcal{M}^k\}$, such that $\mathcal{M}^k \doteq \langle \mathcal{S}_k, \mathcal{A}, \mathcal{R}, p \rangle$. In particular, for a given $\mathcal{S}_k$, the red world state reward distribution is characterized as a Gaussian distribution with mean, $\mu_{\text{red}}$, and standard deviation, $\sigma_{\text{red}}$. Conversely, the blue world state reward distribution is characterized as a mixture of two Gaussian distributions with means, $\mu_{\text{blue-a}}$ and $\mu_{\text{blue-b}}$, standard deviations, $\sigma_{\text{blue-a}}$ and $\sigma_{\text{blue-b}}$, and a mixing coefficient of 0.5.

In the experiment performed, we set $k \in \{1, 2, 3\}$. For all $k$ and all states, we set the standard deviation to 0.05. For $k = 1$, we set $\mu_{\text{red}} = -0.7$, $\mu_{\text{blue-a}} = -1.0$, and $\mu_{\text{blue-b}} = -0.2$. For $k = 2$, we set $\mu_{\text{red}} = -1.5$, $\mu_{\text{blue-a}} = -1.25$, and $\mu_{\text{blue-b}} = -1.0$. For $k = 3$, we set $\mu_{\text{red}} = -0.5$, $\mu_{\text{blue-a}} = -0.9$, and $\mu_{\text{blue-b}} = -0.5$. The indexing function, $\omega$, was defined such that $\omega(t) = 1$ for $t < 40,000$, $\omega(t) = 2$ for $40,000 \leq t < 80,000$, and $\omega(t) = 3$ otherwise.

In terms of the hyperparameters used with the RED CVaR Q-learning algorithm, we used the tuned hyperparameters from Rojas and Lee (2025). That is, $\alpha = 2e\text{-}2$, $\alpha_{\text{CVaR}} \doteq \eta_{\text{CVaR}}\alpha$, where $\eta_{\text{CVaR}} = 1e\text{-}1$, and $\alpha_{\text{VaR}} \doteq \eta_{\text{VaR}}\alpha$, where $\eta_{\text{VaR}} = 1e\text{-}1$. We used a fixed CVaR parameter, $\tau$, of 0.25, an $\varepsilon$-greedy policy with a fixed epsilon of 0.1, and set all initial guesses to zero. The results for this $\mathcal{S}$-*RPBP* task are shown in Figure 4.

