# OpenReview forum: "Ergodic Risk Measures: Towards a Risk-Aware Foundation for Continual Reinforcement Learning"
_ICLR.cc/2026/Conference — ICLR 2026 Conference Withdrawn Submission_

### Official Review · Reviewer_6n6g · 2025-10-27

**Soundness:** 2
**Presentation:** 2
**Contribution:** 2
**Rating:** 2
**Confidence:** 3

**Summary:**

This paper presents a study of risk and its role in continual reinforcement learning. The work is motivated by an argument that there is a necessary link between risk and continual learning agents: if an agent needs to survive indefinitely (a proposed prerequisite for continual learning), then the agent must be-risk aware. With this motivation in mind, the paper argues that existing notions of risk are ill-fit to accommodate the continual RL problem setting. Of special interest is the stability-plasticity dilemma, one of the central challenges facing continual learning agents. That is, an agent must be suitably capable of retaining what it has learned from the past (stability), while also be sufficiently adaptive to any new information (plasticity). The core argument of the paper comes in section 4.1 through two definitions of risk measures (Def 4.1 and Def 4.2), and two subsequent lemmas (Lemma 4.1, Lemma 4.2), that together present an argument that standard risk-measures are not well-suited to extend to continual RL. In light of this proposed shortcoming, the paper then proposes "ergodic risk measures" (Def 4.4) and argues this is one correct way to conceive of risk in continual RL. Then, in section 5 the work presents a case study of this risk measure using CVaR.

**Strengths:**

- Goals, risk, and continual learning are extremely fundamental topics. As such, scrutinising them and their relationship is admirable, given the potential for long-lasting benefit that can come from clarifying the necessary relationship between say risk-awareness and continual learning.

- The approach to invoke axioms and desiderata for thinking carefully about how risk should be conceived of in continual RL is a powerful framing, and I believe has a high potential to contribute something fundamental to the field.

**Weaknesses:**

I believe the paper is asking a deep and important question, but the actual development of the answer to that question leaves a great deal of conceptual confusion, and lacks bedrock justification for the many choices that go into the core arguments. This, to me, the central weakness of the paper.

For the sake of clarity, I perceive the core reasoning of the paper to be as follows:
1. Continual learning implies survival.
2. Survival requires risk-awareness.
3. Thus, risk-awareness is essential to continual RL.
4. However, standard notions (from, say, non-continual RL) are not fit for continual RL, because of their incompatibility with the stability-plasticity dilemma.
5. Therefore, we require a new notion of risk that _is_ fit for continual RL, which motivates the design of Definition 4.4.
6. Definition 4.4 is fit for continual RL.

If the authors disagree with my summary, please feel free to correct me, and we can discuss during the rebuttal period.

___[Points 1-3]___ Assuming that I have well-understood the logical flow of the work, let's dive in to the key pieces. While I find points (1) and (2) open for debate, I do not believe they are actually critical to the narrative of the present paper. With that in mind, let us take it as granted that (3.) is a reasonable starting point for motivating the work: risk has a long history throughout decision theory, philosophy, and RL, and as such it is useful to recover a suitable account of risk in continual RL.

Under this view, the core of the paper is about demonstrating points (4-6), given (3) is taken as a framing assumption.

___[Points 4-6]___ To me, the biggest weakness of the work is in providing a convincing argument in support of each of 4, 5, and 6. I anticipate my reasoning here will be somewhat lengthy, and I am happy to discuss during the rebuttals as well. Let me say more about why:
- 4. The argument given to show why existing risk measures (such as a "static" risk measure) is presented in Section 4.1. The reasoning is effectively that the stability-plasticity dilemma requires a new kind of risk. However, this reasoning is carried out by proposing two new notions of plasticity that are about risk (Defs 4.1 and 4.2); these definitions are not well-motivated, and may not capture the intuitive spirit of what the literature means about plasticity, and do not obviously make contact with the basic intuition of the concept of plasticity. Lemmas 4.1 and 4.2 are then used to show that standard risk measures fail to meet the criteria of these new kinds of plasticity.

My primary critique of the work is that Definitions 4.1 and 4.2 are not, as of yet, well-motivated. Why should we conceive of plasticity in a way that is _not_ about an organism, agent, or learning system, but is about a risk measure? This seems like a fundamental conceptual confusion. Again I can see an argument relating risk and plasticity, and this argument feels like the essential logical piece that the paper requires to separate standard risk from continual RL. However, this argument is not yet presented. Moreover, the stability-plasticity dilemma emerges in the __context__ of continual learning---the nature of the objective that the learning system pursues gives rise to the dilemma. Here, we see that plasticity is invoked first, then risk is assessed after the fact. Shouldn't the dilemma emerge as a consequence of the objective a learning system is pursuing?

- 5. This is echoing some of my reasoning above: just because one existing kind of risk fails to adhere to the two proposed definitions, it does not mean that an entirely new kind of risk is needed. In general, much more is needed to pull apart, precisely, what desiderata are needed to define a coherent and appropriate kind of risk for continual RL.

- 6. Similar to my reasoning in (5.), I worry that not enough bedrock justification or clarity is offered as to what precisely is sought in terms of an account of risk for continual RL. As such, Definition 4.4 is left as an ad-hoc, so far. Furthermore, given the inherently complex nature of continual RL (the cited work by Ring, Kumar et al., and Abel et al. each invoke an arbitrary, history-based environment, that may not contain any recurrence), I am highly skeptical of invoking Assumptions 4.1 and 4.2 in continual RL to get a notion of risk off the ground.

To summarise: I applaud the authors for exploring such an important and fundamental question. I believe there are many important ingredients to this work, though the core argument in point (4.) misses the mark for me: why would we reject a type of risk on the basis of this kind of definition of plasticity about risk? What is needed is a deeper conceptual analysis of the stability-plasticity dilemma, how it relates to different kind of objects that learning systems may face, and to make the point clearly as to why continual learning necessitates risk measures of a certain form.

Furthermore, it is unclear what, precisely, we are after for a measure of risk in continual RL to begin with, but any measures that invoke unichain and communicating assumptions should invalidate the proposal, as such assumptions are antithetical to the essence of continual RL.

Lastly, I believe the work could better engage with the literature on plasticity: see Section 4 of the cited Kumar et al. paper, or work by Chen et al. (2023), Raghavan and Balaprakash (2021), or recent work by Abel et al. (2025).

References:
- Chen, Q., Shui, C., Han, L., & Marchand, M. (2023). On the stability-plasticity dilemma in continual meta-learning: Theory and algorithm. Advances in Neural Information Processing Systems.
- Raghavan, K., & Balaprakash, P. (2021). Formalizing the generalization-forgetting trade-off in continual learning. Advances in Neural Information Processing Systems.
- Abel, D., Bowling, M., Barreto, A., Dabney, W., Dong, S., Hansen, S., ... & Singh, S. (2025). Plasticity as the Mirror of Empowerment. Advances in Neural Information Processing Systems.


As one short aside, I believe the paper could also do a better job of presenting mathematical definitions. Definition blocks are sometimes nearly eight lines of text. The paper would be much more clear if the needed ingredients were introduced one at a time in an intuitive way so that essential definitions are at most a couple of lines.

**Questions:**

My questions are distillations of the weaknesses tab above.

Q1. Why is it sensible to tether plasticity to a measure of risk, rather than as a property of an agent, learning system, or organism? If an agent could, in principle, "solve" continual RL while avoiding the stability-plasticity dilemma entirely, might this suffice?

Q2. What are the bedrock assumptions or properties we do want out of a measure of risk in continual RL? (Can you provide this argument in the absence of any references to plasticity or stability?) Why should we grant ourselves Assumptions 4.1 and 4.2, when these could arguably invalidate the notion of continual RL to begin with (effectively, if the environment mixes, then we can analyze learners at convergence, but this is precisely not what we want in continual RL).

Q3. Is risk viewed as important in terms of the space of _solutions_? Or is it truly the case that we want to order agents in terms of their risk tolerance? In other words, in the language of the vNM axioms, one way to think about risk is to get rid of the Independence axiom (at a minimum). Why should we abandon the Independence axiom, specifically in continual RL?

I realize I am asking quite challenging questions. To give a full account of all of these is probably beyond the scope of a conference paper. However, for the results to stand as valuable to the community, I do believe we need at least partial clarity on the above.

---

> ### Author Response · Authors · 2025-11-20
> **Rebuttal by Authors (1/2)**
>
> We thank the reviewer for having put so much effort and thought into the review of our paper. In particular, the suggested exercise about considering plasticity from the point of view of the agent was particularly fruitful and has subsequently shaped the updated draft of the paper.
>
> In particular, instead of framing plasticity from the perspective of the objective, we now utilize an existing definition of plasticity that frames it from the perspective of the agent [1]. In doing so, we are able to generalize the continual RL framework from [1] into the risk-aware case and use this resulting framework to propose a set of axioms that establish the conditions needed for risk awareness in the continual setting from the perspective of the agent.
>
> We then use these axioms to show why static and nested risk measures are inadequate, as well as to show why our proposed class of ergodic risk measures is suitable for continual RL.
>
> We believe that these changes address the reviewer’s primary concern, in the sense that the axioms now provide sufficiently strong motivation as to why static and nested risk measures are not adequate, and why ergodic risk measures are adequate.
>
> We would be grateful for further discussion with the reviewer if we have misunderstood their primary concern, or if there are additional concerns.
>
> We now address some specific points not covered above:
>
> **Any measures that invoke unichain and communicating assumptions should invalidate the proposal, as such assumptions are antithetical to the essence of continual RL**:
>
> This is addressed in our overall response.
>
> **The work could better engage with the literature on plasticity:**
>
> In addition to now using the framework from [1] as our basis, we now reference the other literature related to plasticity in the updated draft.
>
> **The paper would be much more clear if the needed ingredients were introduced one at a time in an intuitive way so that essential definitions are at most a couple of lines.**
>
> We have incorporated this suggestion into the updated draft of the paper.
>
> **Why is it sensible to tether plasticity to a measure of risk, rather than as a property of an agent, learning system, or organism?**
>
> We are in agreement with the reviewer that it is more sensible to tie plasticity to the agent, and have reflected that in the updated draft.
>
> **What are the bedrock assumptions or properties we do want out of a measure of risk in continual RL?**
>
> In the context of the updated draft, we now argue that the risk should be computable based on the agent’s limited memory, and that agent’s risk assessment and subsequent behaviour should be influenced by changes in its risk attitude as well as new observations from the environment infinitely often (see Proposition 4.3 of the updated draft).
>
> **Due to space constraints, we continue our response in the next comment:**

---

> ### Author Response · Authors · 2025-11-20
> **Rebuttal by Authors (2/2)**
>
> **Is risk viewed as important in terms of the space of solutions? Or is it truly the case that we want to order agents in terms of their risk tolerance? In other words, in the language of the vNM axioms, one way to think about risk is to get rid of the Independence axiom (at a minimum). Why should we abandon the Independence axiom, specifically in continual RL?**
>
> We will attempt to think our way through these questions.
>
> We argue that what matters in terms of risk is not the scalar number that an agent optimizes, but rather the resulting behaviour. That is, does the agent exhibit the behaviour that best aligns with its risk tolerance (e.g. does it pick the least riskiest option if it has low risk tolerance?).
>
> Under this argument, the question then becomes, “how can we induce this kind of behaviour in an agent?” Both expected utility theory and the theory of risk measures attempt to encode the notion of risk in an agent, such that if it can accurately assess how risky something is (via a utility function or a risk measure), then based on this information it can make an appropriate decision and thereby exhibit the desired behaviour.
>
> However, this assumes that the agent’s assessment of risk is ‘sensible’ and ‘good’, which suggests that the agent’s ability to exhibit risk-aware behaviour is limited by its capacity to accurately assess how risky something is. This is perhaps one of the reasons that risk measures are more popular than expected utility theory in RL, as they typically allow for more flexible ways to encode risk.
>
> Now consider continual RL, where the agent must now exhibit even more complex risk-based behaviours. For example, instead of the desired behaviour being “pick the least risky option”, the agent now needs to “pick the best option with reference to an ever-changing risk tolerance”. This now puts more pressure on the agent’s risk assessment. Does the agent’s risk assessment account for changing risk tolerances? This again swings in favour of risk measures, as they may be better suited to encode these more complex requirements.
>
> So what are these requirements exactly? That is, what requirements does our risk measure (or utility function) need to meet in order for a risk assessment to be ‘sensible’ (or even possible) in a continual setting? This is a question that we attempt to answer in the updated draft of our paper based on the resource and plasticity requirements of continual agents (see Proposition 4.3).
>
> This now brings us to the independence axiom, which admittedly, we are not as familiar with (expected utility theory is a different paradigm than risk measures) but will attempt to invoke to answer the reviewer’s question: why should we abandon the independence axiom in continual RL? As per the above reasoning, we conjecture that the independence axiom may impose strong limitations on our ability to encode risk, such that the resulting utility function may not be able to accurately capture the desired behaviours (such as those associated with a changing risk tolerance), the non-stationarity of environments, or the agent’s own limitations in a continual RL setting.
>
> We again thank the reviewer for their insightful and thought-provoking review of our work. We are happy to continue discussing with the reviewer on any additional comments that the reviewer has.
>
> **References:**
>
> [1] Abel et al. 2025. “Plasticity as the Mirror of Empowerment.”

---

> > ### Comment · Reviewer_6n6g · 2025-11-23
> > **Re: Rebuttal**
> >
> > I thank the authors for such a thorough response: given the magnitude of the changes to the paper, I will need some time to read things carefully and better understand the new proposal. The main thrust of the changes sound compelling.
> >
> > This is a short note to say I have read the rebuttal, and will be reading things over in more detail before following up.

---

### Official Review · Reviewer_ocLh · 2025-10-31

**Soundness:** 4
**Presentation:** 1
**Contribution:** 2
**Rating:** 2
**Confidence:** 3

**Summary:**

The authors formulated a mathematical framework for risk-aware continual reinforcement learning (RL). They first presented the existing framework for risk-aware episodic RL, which uses either a static or nested risk measure. They then proved that both measures are incompatible with the stability-plasticity dilemma in continual RL, as the static measure never adapts and the nested measure never forgets. In the second part of the paper, they defined the notion of ergodic risk measure desirable for the continual setting, as it satisfies asymptotic plasticity and local time consistency. They further demonstrated that, under two ergodicity-like assumptions, simply replacing the expectation E[.] in the average reward with any risk measure rho[.] gives a risk-aware objective that is an ergodic risk measure. Finally they tested their newly developed framework by letting rho be a popular risk measure called conditional value-at-risk (CVaR). They conducted experiments with a q-learning agent that optimizes the corresponding risk-aware objective in the continual red-pill blue-pill environment. They demonstrated the agent’s adaptive risk awareness by observing the change in its policy when the risk attitude (controlled by a CVaR param) or the reward distribution is changed.

**Strengths:**

This paper formalizes important properties of a risk-aware objective for continual learning, which could be useful for the continual learning community.

The experimental design (with either the risk tolerance or reward changing) was sensible and clearly explained.

The theory is correct.

**Weaknesses:**

The primary weakness is the contrast to the episodic RL setting, rather than the average reward setting. The paper uses the typical assumptions for the average reward (continuing setting) and even uses the algorithm previously developed for risk-aware average reward (by Rojas and Lee, 2025). The same CVaR decomposition as in Rojas and Lee is used. Yet this paper is pitched saying that the risk-aware measures used in RL are not compatible with continual learning (I think what is meant here is that the episodic risk-aware measures are not compatible).

It is possible that the objectives frm Rojas and Lee and this paper are slightly different, though these differences were not discussed (nor was the Rojas and Lee paper cited when the CVaR objective was proposed in the experimental results section). One difference I do note is that the objective uses pi_t, changing with time, rather than pi. However, I am actually not sure if this is an error or intentional. Are all pi_t independently optimized? The LHS of the equation has CVaR_{pi_t}, and then the RHS has pi_t changing with t in the sum.

The primary contribution seems to be in showing which measures do or do not satisfy the plasticity and local consistency properties. It could be more clear to both explain that episodic RL objectives do not satisfy these properties and that this risk-aware average reward objective (that was recently introduced, in Rojas and Lee) does satisfy these properties. Further, you could then highlight the more general class of ergodic risk measures, and potentially provide some useful examples.

To this point, this paper is currently hard to read and it is largely definitional and it can be hard to determine what is new. Are all the definitions new? Which ones are new?

Additionally the properties are very technical and it is hard to gain much intuition. There is intuition given, but at times it is too brief or too high-level. For example, parts of the plasticity definition are clear and others are not. After Definition 4.2 about asymptotic plasticity, the intuition is clear: the risk evaluation should only depend on recent history. Earlier in this section, it is stated: “That is, the risk evaluation at a given time step should depend only on the recent history leading up to that time step, rather than the entire history.” This leads me to think the static risk measures suffer from using the whole history (do they?)
But then in the proof it says: “Conversely, a static risk measure is a single mapping corresponding to a single point in time. That is, it is not time-indexed, and therefore cannot adapt as new information arrives.” If it depends on the whole history, won’t it provide a different evaluation as new information arrives? Providing slightly more detailed descriptions (not just high-level), in plainer language, on these technical definitions would be very helpful.

As a specific actionable suggestion, it would be useful for this section to
(a) better motivate why we need this plasticity property, potentially with an example
(b) give an example of a static risk measure.
It seems, based on the proof, that Lemma 4.1 is a trivial fact that could just be stated about static risk measures. What is more useful here is to help the reader understand why existing measures are ineffective and why it is important to have this property. I remain unsure if these properties are necessary for continual learning (e.g., with changing rewards as in the experiments) or necessary just for the average-reward setting.

Finally, the goal of the experimental results is not fully clear. There is a demonstration that this algorithm can adapt in this environment, by optimizing this average reward objective. It seemed to me that this was the algorithm proposed by Rojas and Lee, but I could be missing something here.

1. Is the algorithm new? If yes, this should be made more clear.

2. Are there any pertinent baselines to include, to highlight why this new risk-aware objective is useful? For example, could static or nested risk measures be included in the experiments as baselines to show explicitly why they do not work in the continual setting?

**Questions:**

Summarizing the questions outlined above:
1. It would be useful to explain more if this is different from the average reward setting, and potentially even comment on why these typical ergodic assumptions are appropriate given the continual learning motivation. Do they match the experiments? If not, then it would also be useful to comment on why this mismatch is ok.

2. Is the objective the same as from Rojas and Lee, and what does it mean to optimize over pi_t?
Is the algorithm new?

3. What are the goals of the experiments?

4. Do the static risk measures suffer from using the whole history?

5. The key novelty seems to be in characterizing risk measures according to the plasticity and temporal coherence criteria. Are these properties only about plasticity-stability, or are they generally necessary to have sensible risk measures for the average reward setting? It would be useful here to give more insight into why we care about these definitions, and also if these risk measures have previously been used for average reward.

---

> ### Author Response · Authors · 2025-11-20
> **Rebuttal by Authors (1/2)**
>
> We thank the reviewer for their thorough and thoughtful review of our paper. We are happy to provide clarifications and comments below:
>
> **It would be useful to explain more if this is different from the average reward setting, and potentially even comment on why these typical ergodic assumptions are appropriate given the continual learning motivation.**
>
> This is addressed in our overall response.
>
> **It is possible that the objectives from Rojas and Lee and this paper are slightly different, though these differences were not discussed**
>
> The objective from [1] is defined for a single average-reward MDP. The objective defined in this paper is defined for an infinite sequence of average-reward MDPs.
>
> Given ergodicity assumptions (which we do assume in this paper), the objective from [1] is independent of initial conditions. Accordingly, the objective considered in our paper effectively remains the same as the objective from [1], with the only difference being that the algorithm starts at a different ‘initial’ state (which doesn’t affect the final solution as per the ergodicity assumptions).
>
> **The Rojas and Lee paper was not cited when the CVaR objective was proposed in the experimental results section**:
>
> We kindly remind the reviewer that [1] was cited in lines 407-408 of the submitted draft after the CVaR objective was introduced (lines 460-463 of the updated draft).
>
> **Are all pi_t independently optimized?**
>
> Correct, there is a policy for each MDP. Once the agent switches into a new MDP it effectively starts optimizing a new policy for that new MDP (see lines 108-125 of the submitted draft; or lines 368-379 of the updated draft).
>
> **The primary contribution seems to be in showing which measures do or do not satisfy the plasticity and local consistency properties. It could be more clear to both explain that episodic RL objectives do not satisfy these properties and that this risk-aware average reward objective (that was recently introduced, in Rojas and Lee) does satisfy these properties. Further, you could then highlight the more general class of ergodic risk measures, and potentially provide some useful examples.**
>
> We have incorporated much of this suggestion into the updated draft of the paper. We are happy to further discuss with the reviewer if there are still concerns in this regard.
>
> **To this point, this paper is currently hard to read and it is largely definitional and it can be hard to determine what is new. Are all the definitions new? Which ones are new?**
>
> We have removed excess definitions in the updated draft of the paper to make it more readable. The primary new definition of this work (both in the submitted and updated draft) is that of ergodic risk measures. In the submitted draft, we also defined the plasticity and local time consistency properties. However, in the updated draft, based on another reviewer’s suggestions, we now use a prior definition of plasticity from [2].
>
> All in all, in the updated draft, the new notions introduced are as follows:
> - Risk-aware observations and risk-aware plasticity in the context of the continual RL framework from [2]
> - The axioms for risk-awareness in the continual setting
> - Definition of an ergodic risk measure
> - The conditions under which the risk-aware average-reward objective is an ergodic risk measure.
>
> Note: we have used formatted boxes to highlight the primary contributions in the updated draft.
>
> **Due to space constraints, we continue our response in the next comment:**

---

> > ### Author Response · Authors · 2025-11-20
> > **Rebuttal by Authors (2/2)**
> >
> > **It would be useful to better motivate why we need this plasticity property and provide better intuition in plain language**:
> >
> > We believe that, now that we are using an existing framework and definition of plasticity (from [2]), that the motivation and intuition is better communicated. We are happy to further discuss with the reviewer if they find that there are still parts of the paper that need stronger motivations or intuitions.
> >
> > **The goal of the experimental results is not fully clear**
> >
> > At a high level, the aim of the experiments is to show that optimizing an ergodic risk measure induces risk-aware behaviour that ‘makes sense’ in a continual setting. This includes adapting from a changing risk attitude and changes in the environment.
> >
> > Accordingly, in this work we do not propose a new algorithm, but rather, what we show is that the algorithm from [1] belongs to a class of methods (i.e. ergodic risk measures) that had not been formalized until now. Moreover, the experiments show that our proposed class of ergodic risk measures can indeed induce the desired risk-aware behaviour.
> >
> > In terms of baselines, given that we rule out static and nested risk measures from a theoretical perspective, we do not believe that adding them as baselines would add much value. Even so, how one can or should make a proper empirical comparison between the various classes of risk measures is not well understood and beyond the scope of this work (i.e., in the literature, static and nested measures are often compared theoretically, but not empirically).
> >
> > **Do the static risk measures suffer from using the whole history?**
> >
> > Yes, as well as the fact that they are not well-defined for infinite horizon settings (i.e. they evaluate risk at a single point in time, like the end of an episode)
> >
> > **Are these properties only about plasticity-stability, or are they generally necessary to have sensible risk measures for the average reward setting?**:
> >
> > The properties establish the conditions needed so that the agent can properly assess risk in the continual setting. The connection with the average-reward setting is that its objective ‘naturally’ satisfies these conditions (this is what we show in the paper).
> >
> > We again thank the reviewer for their insightful review of our work. We are happy to continue discussing with the reviewer on any additional comments that the reviewer has.
> >
> > **References:**
> >
> > [1] Rojas et al. 2025 “Burning RED: Unlocking Subtask-Driven Reinforcement Learning and Risk-Awareness in Average-Reward Markov Decision Processes.”
> >
> > [2] Abel et al. 2025. “Plasticity as the Mirror of Empowerment.”
> >
> > [3] Xia et al. 2023 “Risk‐sensitive Markov Decision Processes with Long‐run CVaR Criterion”

---

> > > ### Comment · Reviewer_ocLh · 2025-11-23
> > > **Follow-up questions about the objectives**
> > >
> > > Thank you for this in-depth response. As another reviewer said, I need a bit more time to absorb these answers and changes.
> > >
> > > But I remain a bit confused about the objective, and hope I can clarify that here. For many of the formulas (e.g., (4), (5), (6)), there is t changing on the RHS and pi_t on the LHS for a fixed t. You clarified that the objective is for every pi_t. However, the math is imprecise. What are the parameters on the LHS? The index t is used for "on this time step t" and the index k is used for the current MDP. Are we trying to learn one stationary policy for each k? Are we trying to learn one policy for each time step, pi_1, pi_2, ....? Ideally, on the LHS, you could write what is being optimized (policy parameters? Or pi_1, pi_2, ...? Or?), and allow t to only be the variable in the sum on the RHS.
> > >
> > > In general, I have a hard time reconciling the objective that is an limiting, average reward objective with the fact that we are switching between MDPs before we reach that limiting behavior. Seeing a more precise objective and why it is reasonable to be able to optimize such an objective would really clarify.

---

> ### Author Response · Authors · 2025-11-23
> **Authors response to follow-up questions**
>
> We thank the reviewer for their timely response. We are happy to provide the requested clarifications:
>
> **I remain a bit confused about the objective, and hope I can clarify that here.**
>
> We thank the reviewer for pointing out the inconsistency in our notation. We have updated Equations 4-6 in the updated draft of the paper to remove the use of $t$ on the RHS of the equations. We have also updated our notation to use $\pi^{k}$ vs $\pi_{t}$ to better distinguish between the two.
>
> For clarity, we are trying to learn one stationary policy for each $k$ (not one policy for each time step $t$). The notation $\pi_t$ denotes which policy is being followed at time $t$ (i.e. $\pi_{t} \in$ {$\{\pi^{k}\}$} ). We note that we define an indexing function $\omega(t)$ (see lines 370-373) that tracks which of the policies in {$\{\pi^{k}\}$} is followed at time $t$.
>
> We have opted to keep $t$ on the LHS as we believe that it is important to emphasize the time-dependency of the objective. Of course, if the reviewer disagrees with this decision, we are happy to further discuss.
>
> In words, each objective in Equations 4-6 is as follows: when the agent switches from MDP $M^{k}$ to $M^{k+1}$, it now seeks to optimize the long-run {average reward, risk measure, CVaR} of MDP $M^{k+1}$ via policy $\pi^{k+1}$. Here, $h_{0}$ (formerly $t_{0}$) is defined as the time step where the change from $M^k$ to $M^{k+1}$ occurs (see line 378).
>
> **In general, I have a hard time reconciling the objective that is limiting with the fact that we are switching between MDPs before we reach that limiting behavior. Seeing a more precise objective and why it is reasonable to be able to optimize such an objective would really clarify.**
>
> We are happy to provide some comments in this regard:
>
> As correctly pointed out by the reviewer, there may indeed be a switch of MDPs before the limiting behaviour is reached. We will address this case below, but before we do, we want to point out that it is still very much possible that the limiting behaviour *is* reached before the switch in MDPs happens (such as in the experiments provided in the paper). As such, while we do not disagree with the reviewer on this point, we do want to clarify that it is not automatically the case that the limiting behaviour is never reached for any MDP.
>
> Now, what if the limiting behaviour is not reached before the switch in MDPs happens? We claim that this is not problematic. To see why, we will need to go into a bit of detail with how average-reward RL works in our explanation but we will try to keep it brief:
>
> First, we note that given the ergodicity-like assumptions, the objective remains well-defined regardless of whether the limiting behaviour is reached or not. That is, there always exists a unique optimal average-reward (or risk measure / CVaR) for each MDP that the agent could “find” (more precisely, it could find a policy that yields this optimal average-reward) if given enough time. As such, from a theoretical perspective, there is no issue with a pre-emptive change in MDP; all it means is that the agent did find the optimal solution in the allotted time that the specific MDP was “active”.
>
> Now, just because the agent does not find the optimal solution, it does not mean that it makes no progress at all. As an analogy, consider any deep RL agent. It is widely accepted that deep RL agents are very unlikely to find the true optimal solution, however, this does not stop them from achieving reasonable behaviours. That is, in search of optimality, these agents are still able to achieve desired behaviours despite not actually reaching optimality.
>
> A similar story applies in the case of a pre-emptive MDP switch with an average-reward-like objective. In particular, in any average-reward RL algorithm, the agent keeps an estimate of the objective (e.g. an estimate of the limiting average-reward) and updates this estimate based on new observations (similar to a value function). Hence, the agent will always have an estimate of what the optimal limiting {average-reward, risk measure, CVaR} is, and use that estimate to direct its behaviour. Importantly, the quality of this estimate (and the subsequent behaviour) continually improves over time, regardless of whether the agent actually reaches the limiting behaviour or not (similar to how the quality of the value function estimates in a deep RL agent will get better over time regardless of whether it finds the optimal solution).
>
> As such, both from a theoretical and empirical perspective, we do not see an inherent issue with an MDP switch before the limiting behaviour is reached (of course, one could argue “what if the MDP switches at every time step etc.” however, such a case would clearly cause trouble for any objective, so we do not view it as a specific issue with our objective).
>
> We appreciate the reviewer's time and look forward to further discussions with the reviewer.

---

### Official Review · Reviewer_7NtN · 2025-11-01

**Soundness:** 2
**Presentation:** 3
**Contribution:** 2
**Rating:** 2
**Confidence:** 3

**Summary:**

The submission provides an attempt to formalize the problem setting of risk-aware continual reinforcement learning. The authors give a definitions for properties of conditional risk measures they argue are appropriate for continual learning setting and provide an objective that satisfies those properties.  An experiment shown demonstrating that a RED CVaR Q-Learning agent can learn in an environments with risk preferences and reward functions that change.

**Strengths:**

The introduction motivates the problem well. The paper is ambitious in that it is the first attempt at formalizing risk-aware RL for the continual learning.

**Weaknesses:**

Section 3.1 Regarding the definition of continual learning.  It is important to acknowledge that there is not a widely accepted definition of continual reinforcement learning. To my knowledge, the most popular definition was provided by Abel et al. 2023 (which is cited in this text but not discussed). The notation given here frames continual reinforcement learning as a sequence of MDPs indexed by a function of time. This is certainly a possible definition, although, it seems more appropriate to call it a definition of non-stationary RL.  Allowing the sequence of MDPs to be infinite where they "may differ in its state-space" certainly does lend itself to requiring continual learning (as per Abel et al. 2023).  However the definition of an infinite sequence of MDPs is not used again and the rest of the paper assumes a single MDP!  This disconnect makes it difficult to see how the prescribed properties and objective relates to the continual learning setting as proposed.

(Definition 4.2 Asymptotic Plasticity) is a weak property. The description in the text describing Definition 4.2 in English does not match how it is defined. The sentence says "...the influence of any past history vanishes, such that the risk evaluations effectively depend only on the more recent history". This is a claim about any finite length of past history. That is, for all $n$, in the limit of infinite time, the influence of the past history vanishes. But this is not what the definition says.  The definition says there exists finite $n$ such that the risk measure does not depend on the history prior to time step $n$. The only things we are given is that $n$ exists and is finite. What if $n=1$? Then the plasticity property is saying the influence of only the first timestep in history is forgotten and all other history matters?

(Definition 4.4 Local Time Consistency) I have a similar concern as above with respect to the logical quantifiers in definition 4.4. The statement is there exists an $n$ and an $m$ such that time consistency holds on the interval $n \leq t \leq n+m$. What if there is one $n$ and $m=1$, then we only require that the risk metric is consistent for one time step? I would be more interested if the definition was something like for all $n$ there exists a finite $m$ such that time consistency holds on the interval $n \leq t \leq n+m$.

(Equation 2) The common objective in RL is to maximize expected return, that is, the sum of rewards. Of course because expectation is linear, one can put the expectation inside the sum and write it as sum of expected rewards. This however, does not hold for general risk measures. It is strange that the objective in equation 2 is written as the sum of risk-measures of rewards instead of risk-measure of the sum of rewards.

Also, I believe there is a typo in the policy subscript of Equation (2). I assume it is supposed to be $\rho_{\pi_{t_0}}$  and not $\rho_{\pi_{t}}$ as is written. Right now, $t_0$ is an unbound variable. The same typo appears in the proof.

(Assumptions 4.1 and 4.2) Both assumptions are making assumptions about a single MDP, breaking away from the continual learning setting given in section 3.1.

Overall it's not clear that the claims from the abstract are adequately supported with evidence.

The abstract states that "we show that the classical theory of risk measures, widely used as a theoretical foundation in non-continual risk-aware RL, is, in its current form, incompatible with the continual setting".  The main evidence is that the authors present a particular property (asymptotic plasticity) and then claim that a particular choice of nested risk measures doesn't satisfy this property (or time consistency).  The argument for why these properties are necessary is not clear.  Or why these properties are sufficient either.  Furthermore, the author's proposed objective seems to rely on classical risk measures with $\rho$.  And it seems you could even generalize it to allow $\rho_t$ or $\rho_{t_0}$.  All of this is to say that the word incompatibility suggests to me a much stronger claim than what's supported.

The abstract also states that "We extend risk measure theory into the continual setting by introducing a new class of ergodic risk measures that are compatible with continual learning".  None of this feels like an extension of risk measure theory.  It's then not clear what problem the class of ergodic risk measures is solving, or in what way that class would uniquely be compatible.

Furthermore, it's unclear exactly what the experiments are trying to show (see Question below).

**Questions:**

I'm somewhat confused what the experiments are trying to show. The experiments provided are non-stationary variants of the red-pill blue-pill experiments in Rojas and Lee (2025).  In that work, convergence was proved for tabular RED CVaR Q-Learning and empirically demonstrated within 4k time steps (See Figure 3 of Rojas and Lee (2025)). In these experiments, they are run for 100k time steps. Is the point to demonstrate that RED CVaR Q-Learning can recover from changing in risk tolerance? I'd appreciate if the authors can comment on what was the intention here as the experiments don't seem fit with the rest of the paper.

---

> ### Author Response · Authors · 2025-11-20
> **Rebuttal by Authors**
>
> We thank the reviewer for their thorough and thoughtful review of our paper. We are happy to provide clarifications and comments below:
>
> **The definition of an infinite sequence of MDPs is not used again and the rest of the paper assumes a single MDP**
>
> We kindly remind the reviewer that we do not assume a single MDP at any point in the paper. For example, the risk-aware RL objective (Eq 2 in the submitted draft, Eq 5 in the updated draft) is defined for an infinite sequence of MDPs (it is just a risk-aware generalization of the regular objective that was defined for the infinite sequence of MDPs). Similarly, the CVaR objective is defined for an infinite sequence of MDPs, and it was mentioned in lines 419-420 and 422-426 of the submitted draft that the experiments considered involve multiple MDPs.
>
> **Asymptotic Plasticity and Local Time Consistency: What happens if n=1 or m=1 etc.?**
>
> In the updated draft, we have removed these definitions and used an existing definition of plasticity from [1]. Our updated definition of ergodic risk measures (Definition 4.1) does have similar wording to that of the original Asymptotic Plasticity, however we have clarified the wording and notation based on the reviewer’s comments.
>
> With reference to the submitted draft, our aim was (and continues to be) to provide the weakest (i.e. most generic) set of properties required to define an appropriate risk measure for the continual setting. The aim was not to tackle the question of “what is the optimal amount of plasticity or local time consistency” but rather establish the formalism through which such questions could be addressed in the future.
>
> **It is strange that the objective in equation 2 is written as the sum of risk-measures of rewards instead of risk-measure of the sum of rewards.**
>
> This is a risk-aware generalization of the standard average-reward objective (e.g. see Eq 10.6 of Sutton & Barto). This is a purposeful design choice and follows prior works that have argued that the average-reward formulation is a natural fit for continual learning (e.g. [2]). We note that this was mentioned in lines 124-129 of the submitted draft.
>
> **$t_0$ is an unbound variable. The same typo appears in the proof.**
>
> $t_0$ was defined in line 123 of the submitted draft (line 378 of the updated draft; note that this variable has been renamed to $h_0$).
>
> **Overall it's not clear that the claims from the abstract are adequately supported.**
>
> In the updated draft, we address this by using an existing framework for continual RL [1] to derive axioms that establish the minimum conditions necessary for risk-awareness in the continual setting. Then based on these axioms, we show that none of the existing classes of risk measures (i.e., risk measure theory, in its current form) satisfy all the axioms. Finally, we derive a new class of risk measures (an extension of risk measure theory) that does satisfy all axioms and show empirically that it provides a way to do sensible risk-aware learning in a continual setting.
>
> **It is unclear exactly what the experiments are trying to show. Is the point to demonstrate that RED CVaR Q-Learning can recover from changing in risk tolerance?**
>
> At a high level, the aim of the experiments is to show that optimizing an ergodic risk measure induces risk-aware behaviour that ‘makes sense’ in a continual setting. This includes ‘recovering’ from a changing risk attitude.
>
> Now, one may ask “what makes this so significant?” Well, consider the alternative. If we did not have an algorithm that can optimize an ergodic risk measure then we would need to use an algorithm that optimizes a static risk measure or a nested risk measure. An algorithm that optimizes a static risk measure would always be influenced by the prior time steps, such that in the event of the agent’s risk attitude changing, it would likely settle on some policy that changes between the two states, rather than staying in whichever state has less risk. Similarly, a nested risk measure, by definition, cannot change its risk preferences over time, so it would always need to stay in the same state, even if the risk attitude changed.
>
> In fact, to the best of our knowledge, our work is the first to offer a sufficiently flexible framework that can enable (both theoretically and empirically) a scenario where the agent’s risk attitude changes.
>
> We have updated the wording in section 5 (see the updated draft) to make the motivation/intent behind the experiments more clear.
>
> We again thank the reviewer for their insightful review of our work. We are happy to continue discussing with the reviewer on any additional comments that the reviewer has.
>
> **References:**
>
> [1] Abel et al. 2025. “Plasticity as the Mirror of Empowerment.”
>
> [2] Kumar et al. 2025 “Continual Learning as Computationally Constrained Reinforcement Learning.”

---

> > ### Comment · Reviewer_7NtN · 2025-11-26
> > **This Seems to be New Paper**
> >
> > The authors have replaced the entire definitional foundation of their theoretical work in their revision.  This may very well be an improvement.  I simply can't say with confidence (although see below).  However, it is a different paper making different, even if related, claims.
> >
> > I don't believe this is the intent of the revision mechanism of ICLR.  Nor is it reasonable to expect reviewers to review a second paper.  The appropriate course of action is to start the review process again next year or at another conference with this new framing.
> >
> > The authors comments do offer some explanation for confusions I noted.  However, I don't find most of them convincing.
> >
> > I don't even know what to say about the time-sequenced MDPs.  The new revision now introduces a wild mismatch between the Abel (2025) framework (that they use to define axioms) and Section 4.3.1.  Abel uses a history process framework for describing interaction, which does not make ergodicity/unichain/communicating assumptions.  Meanwhile, the treatment in the revision introduces an MDP, and explicitly makes (Assumptions 4.2 and 4.3) ergodicity assumptions.  Elelimy et al. (RLC 2025) make the argument that MDPs (explicitly due to ergodicity assumptions) are ill-suited for thinking about continual RL.  Assumptions that guarantee the existence of an average reward are antithetical to the typical definitions of continual RL (e.g., Abel et al., 2023).  Maybe the sequence of MDPs elides the concern.  But this only raises an important question of why must the limit of Equation (5) even exist.  I don't think without further assumptions it does (and this has nothing to do with risk-awareness as it may not exist in the risk-neutral setting; e.g., consider oscillating between two MDPs with exponentially increasing durations where one MDP has only rewards of +1 and the other only rewards of -1.  In such a case, no average reward exists.  The paper has no mention of aperiodic in the "ergodicity-like" assumptions, which may also be problematic.)
> >
> > I also believe more needs to be defended around applying risk only to one step rewards.  While this may be a treatment that others have made for risk-aware RL in an average reward treatment.  It seems far from the only one.  For example, Wang and Delage (2025) don't seem to arrive at this oversimplification.  Similarly, why not move the $\frac{1}{h}\sum$ inside the CVAR, leaving only the limit outside.  By forcing all risk-awareness into an instantaneous performance measure where the goal is then to optimize the long-range average you are basically reducing risk-awareness in continual RL to simple continual RL again (with a transformation of the reward), which basically makes the whole enterprise pretty trivial.  This seems problematic for a paper proposing to build a risk-aware foundation for continual RL.

---

> ### Author Response · Authors · 2025-11-27
> **Response to Reviewer Comments (1/2)**
>
> We thank the reviewer for their most recent comments. Below is our response:
>
> **I don't believe this is the intent of the revision mechanism of ICLR. Nor is it reasonable to expect reviewers to review a second paper.**
>
> If the reviewer does not wish to engage with the updated draft they are well within their rights to do so.
>
> **The author's comments do offer some explanation for confusions I noted. However, I don't find most of them convincing.**
>
> We are happy to provide more detailed justifications if the reviewer can articulate what aspects they do not find convincing.
>
> **The new revision now introduces a wild mismatch between the Abel (2025) framework (that they use to define axioms) and Section 4.3.1. Abel uses a history process framework for describing interaction, which does not make ergodicity assumptions. Meanwhile, the treatment in the revision introduces an MDP, and explicitly makes (Assumptions 4.2 and 4.3) ergodicity assumptions.**
>
> We kindly note that the Abel (2025) framework is a generic framework, such that an MDP, with or without ergodicity assumptions, can be viewed as a special instance of that framework.
>
> Now, in the updated draft of our work, which the reviewer is clearly referencing in this comment, we extend the Abel (2025) framework to the risk-aware case without invoking MDPs or ergodicity assumptions. That is, all results prior to Section 4.3.1 do not require MDPs or make ergodicity assumptions. Then, starting in Section 4.3.1, we show a specific *instance* of our risk-aware framework that does use MDPs and ergodicity assumptions. In doing so, we are able to provide first-of-its-kind empirical results that show risk-aware decision-making in a continual setting (in a way that is fully-compliant with Abel 2025).
>
> All in all, the claim that there is a mismatch between Section 4.3.1 and Abel 2025 is clearly incorrect, given that Abel 2025 is meant to serve as a generic framework, of which the results in Section 4.3.1 are a special case of.
>
> **Elelimy et al. (RLC 2025) make the argument that MDPs (explicitly due to ergodicity assumptions) are ill-suited for thinking about continual RL. Assumptions that guarantee the existence of an average reward are antithetical to the typical definitions of continual RL (e.g., Abel et al., 2023). Maybe the sequence of MDPs elides the concern. But this only raises an important question of why must the limit of Equation (5) even exist. I don't think without further assumptions it does.**
>
> As mentioned above, in the updated draft of the paper, we propose a risk-aware extension of Abel 2025 that does not invoke the use of MDPs or ergodicity assumptions. The MDP-based case study is a specific instance of our framework, which we utilize to provide tangible empirical results.
>
> Moreover, as hinted by the reviewer, the use of an infinite sequence of MDPs (rather than a single MDP, as in the non-continual case) mitigates the concerns related to ergodicity, since the assumptions need only apply for each MDP independently (see our overall rebuttal for a more detailed answer related to ergodicity and continual RL). Accordingly, this allows us to define a well-defined objective that is in-line with Abel 2023. More concretely, since the objective is time-dependent, in the language of Abel 2023, the agent never remains at a basis since the objective will change when the MDP that the agent is in changes.
>
> Indeed, the ergodicity assumptions are required for the the objective to *exist* (i.e. be well-defined) but as detailed in our response to reviewer ocLH, the important thing is that the objective exists, and not whether the agent actually reaches it.
>
> Finally, we kindly point out to the reviewer that there are several works, such as Kumar 2025, that have argued that the average-reward objective is a suitable objective for continual RL.
>
> **The paper has no mention of aperiodic in the "ergodicity-like" assumptions, which may also be problematic.**
>
> We use the standard unichain and communicating assumptions which have been shown to not be problematic in several prior average-reward RL works, such as [1].
>
> **Due to space constraints we continue our response in the next comment:**

---

> > ### Author Response · Authors · 2025-11-27
> > **Response to Reviewer Comments (2/2)**
> >
> > **I also believe more needs to be defended around applying risk only to one step rewards. While this may be a treatment that others have made for risk-aware RL in an average reward treatment. It seems far from the only one. For example, Wang and Delage (2025) don't seem to arrive at this oversimplification.**
> >
> > This is precisely one of the key contributions of our paper. In particular, we show, in Lemma 4.2, that nested risk measures, which are the ones used in Wang and Delage (2025), are not compatible with continual RL, since they assume that the agent never changes its mind about how risky one outcome is relative to other outcomes, which implies that the agent would not have any plasticity. Conversely, we rigorously show that our proposed class of risk measures, which include risk measures applied to the one-step rewards, is compatible with continual RL.
> >
> > Moreover, we also kindly remind the reviewer that the entire premise of the average-reward setting is centered around the performance of the one-step rewards (i.e., the long-run average of the one-step rewards). Accordingly, applying risk to the one-step rewards is fully in-line with the ethos of the average-reward setting (we note that this was argued rigorously in [2]).
> >
> > **Similarly, why not move the sum inside the CVAR, leaving only the limit outside.**
> >
> > We are simply taking a standard RL objective (in this case, the average-reward objective as defined in Equation 10.6 of Sutton & Barto) and changing the expectation to a risk measure. This is the standard procedure done to convert a risk neutral objective to a risk-aware objective.
> >
> > **By forcing all risk-awareness into an instantaneous performance measure where the goal is then to optimize the long-range average you are basically reducing risk-awareness in continual RL to simple continual RL again (with a transformation of the reward), which basically makes the whole enterprise pretty trivial. This seems problematic for a paper proposing to build a risk-aware foundation for continual RL.**
> >
> > From a purely technical perspective, this claim is not correct. We are happy to provide clarifications:
> >
> > First, we note that changing the expectation in a standard RL objective to a risk measure, such as CVaR, is, by definition, risk-aware RL (this is exactly what we do in the paper). If the claim that the reviewer is making was true, then all prior risk-aware RL works, including Wang and Delage (2025) which the reviewer mentions, could be trivialized as “simple” RL works with a modified reward. Yet, this is generally not considered to be the case because optimizing risk measures is typically a much more difficult process than optimizing expectations.
> >
> > This brings us to our second point: it has been widely shown in the literature that risk-aware RL is a much tougher problem to solve than regular RL. We can appreciate that in our paper this may not appear to be the case since we simply use a prior CVaR algorithm [3] without going into detail about how it works, however, the algorithm itself is highly non-trivial, as are all risk-aware RL algorithms.
> >
> > In other words, optimizing the long-run average CVaR is not the same as optimizing the long-run average average-reward (since both can yield different optimal policies; see our first experiment). In addition, optimizing CVaR (and most risk measures) is a much more difficult problem to solve than optimizing the average-reward, and requires more advanced methodologies to do so.
> >
> > We look forward to further discussions with the reviewer.
> >
> > **References:**
> >
> > [1] Wan et al. 2021 “Learning and Planning in Average-Reward Markov Decision Processes.”
> >
> > [2] Xia et al. 2023 “Risk‐sensitive Markov Decision Processes with Long‐run CVaR Criterion”
> >
> > [3] Rojas et al. 2025 “Burning RED: Unlocking Subtask-Driven Reinforcement Learning and Risk-Awareness in Average-Reward Markov Decision Processes.”

---

### Official Review · Reviewer_avn6 · 2025-11-12

**Soundness:** 2
**Presentation:** 2
**Contribution:** 2
**Rating:** 2
**Confidence:** 4

**Summary:**

The paper studies risk-aware continual reinforcement learning (RL), which addresses lifelong learning with endless adaptation and balances stability and plasticity. Existing risk-aware RL methods are risk-neutral or static/nested risk measures, which are incompatible with continual RL due to the stability-plasticity dilemma. This paper introduces ergodic risk measures, a new class of non-nested dynamic risk measures satisfying asymptotic plasticity and local time consistency, making them suitable for continual RL. The authors theoretically show the compatibility of ergodic risk measures with continuous settings under ergodicity assumptions, and then empirically validated the proposed risk measures by optimizing Conditional Value-at-Risk (CVaR) using RED CVaR Q-learning in continual learning.

**Strengths:**

**The following are the strengths of the paper:**
1. This paper considers the risk-aware continual reinforcement learning and proposes an episodic risk measure for continual RL that addresses the stability-plasticity dilemma.

2. The authors show that the proposed episodic risk measures satisfy asymptotic plasticity and local time consistency.

3. Finally, the authors empirically validate the proposed risk measures by optimizing Conditional Value-at-Risk (CVaR) using RED CVaR Q-learning in continual learning.

**Weaknesses:**

**The following are the weaknesses of the paper:**
1. The ergodicity-like assumptions (unichain and communicating) can be strong and may limit applicability in non-ergodic real-world applications, e.g., when RL dynamics may be changing.

2. The overall experiments are limited, as the authors have tested only two variations of a toy red-pill blue-pill continual learning task.

3. As existing risk-aware RL works also give theoretical results (regret minimization, convergence) to show the optimality of their proposed methods, this paper does not consider the optimality of the proposed method.

4. It is unclear why the risk defined in Eq. 7 is selected, as it may not be able to capture long-term dependencies (across states). Considering non-nested risk measures may not be able to capture the case where the risk will propagate through the states.

**Questions:**

Please address the weaknesses of the paper. I have a few more questions/comments:
1. Define the stability-plasticity dilemma upfront in the paper.

2. Line 326: What does possibly-coherent risk measure mean?

3. How does ergodic risk measure behave and adapt in highly non-stationary or non-ergodic environments where assumptions do not hold? How can it be extended using function approximation or deep RL architectures?

**Details Of Ethics Concerns:**

I do not find any ethical concerns.

---

> ### Author Response · Authors · 2025-11-20
> **Duplicated Review**
>
> Dear Area Chair,
>
> This seems to be a bug with OpenReview, as this is a duplicate review of the one submitted by Reviewer avn6 (note that we have addressed the comments of these reviews under the other version of this review).
>
> Thanks,
>
> Authors

---

### Official Review · Reviewer_avn6 · 2025-11-12

**Soundness:** 2
**Presentation:** 2
**Contribution:** 2
**Rating:** 2
**Confidence:** 4

**Summary:**

The paper studies risk-aware continual reinforcement learning (RL), which addresses lifelong learning with endless adaptation and balances stability and plasticity. Existing risk-aware RL methods are risk-neutral or static/nested risk measures, which are incompatible with continual RL due to the stability-plasticity dilemma. This paper introduces ergodic risk measures, a new class of non-nested dynamic risk measures satisfying asymptotic plasticity and local time consistency, making them suitable for continual RL. The authors theoretically show the compatibility of ergodic risk measures with continuous settings under ergodicity assumptions, and then empirically validated the proposed risk measures by optimizing Conditional Value-at-Risk (CVaR) using RED CVaR Q-learning in continual learning.

**Strengths:**

**The following are the strengths of the paper:**
1. This paper considers the risk-aware continual reinforcement learning and proposes an episodic risk measure for continual RL that addresses the stability-plasticity dilemma.

2. The authors show that the proposed episodic risk measures satisfy asymptotic plasticity and local time consistency.

3. Finally, the authors empirically validate the proposed risk measures by optimizing Conditional Value-at-Risk (CVaR) using RED CVaR Q-learning in continual learning.

**Weaknesses:**

**The following are the weaknesses of the paper:**
1. The ergodicity-like assumptions (unichain and communicating) can be strong and may limit applicability in non-ergodic real-world applications, e.g., when RL dynamics may be changing.

2. The overall experiments are limited, as the authors have tested only two variations of a toy red-pill blue-pill continual learning task.

3. As existing risk-aware RL works also give theoretical results (regret minimization, convergence) to show the optimality of their proposed methods, this paper does not consider the optimality of the proposed method.

4. It is unclear why the risk defined in Eq. 7 is selected, as it may not be able to capture long-term dependencies (across states). Considering non-nested risk measures may not be able to capture the case where the risk will propagate through the states.

**Questions:**

Please address the weaknesses of the paper. I have a few more questions/comments:
1. Define the stability-plasticity dilemma upfront in the paper.

2. Line 326: What does possibly-coherent risk measure mean?

3. How does ergodic risk measure behave and adapt in highly non-stationary or non-ergodic environments where assumptions do not hold? How can it be extended using function approximation or deep RL architectures?

**Details Of Ethics Concerns:**

I do not find any ethical concerns.

---

> ### Author Response · Authors · 2025-11-20
> **Rebuttal by Authors**
>
> We thank the reviewer for their review of our work. Below we address the concerns brought up by the reviewer as well as questions asked by the reviewer:
>
> **The ergodicity-like assumptions can be strong and may limit applicability in non-ergodic real-world applications:**
>
> This is addressed in our overall response.
>
> **The overall experiments are limited.**
>
> We do not disagree, however we note that the amount of experiments conducted is in line with similar works done on continual RL, such as [1, 2].
>
> **As existing risk-aware RL works also give theoretical results (regret minimization, convergence) to show the optimality of their proposed methods, this paper does not consider the optimality of the proposed method.**
>
> Existing works on risk-aware RL that show convergence or regret minimization already have an existing class of risk measures (static or nested) that can be used as a starting point. By contrast, in this work we show that in continual RL we cannot use either class (see Lemmas 4.1, 4.2 of our paper). Accordingly, the focus of our paper is to rigorously motivate and derive an appropriate class of risk measures that is compatible with the continual setting. Additionally, in our case study we show that an existing RL objective with pre-existing convergence guarantees [3] is an instance of an ergodic risk measure, thereby showing that it is possible to derive convergence guarantees for ergodic risk measures. Finally, we note that showing the convergence of risk measures is heavily dependent on the specific risk measure used, whereas the theoretical focus of our work considers a generic class of risk measures.
>
> **It is unclear why the risk defined in Eq. 7 is selected, as it may not be able to capture long-term dependencies (across states). Considering non-nested risk measures may not be able to capture the case where the risk will propagate through the states.**
>
> As per Lemma 4.2, we show that we cannot use a nested risk measure in the continual setting. Moreover, we disagree with the reviewer that the risk defined in Eq 7 (Eq 6 in the updated draft) cannot capture long-term dependencies across states. In fact, optimizing this objective requires that the agent learns which states to visit infinitely often in order to minimize the risk (see [3, 4]).
>
> **Define the stability-plasticity dilemma upfront in the paper.**
>
> This has been addressed in the new draft in Section 3.1, where we invoke the definition of plasticity from [2] to define the dilemma.
>
> **Line 326: What does possibly-coherent risk measure mean?**
>
> It means that the risk measure can also be (but does not have to be) a coherent risk measure, as defined in Definition A.1. In the updated draft, we have removed references to coherent risk measures in the main body as it is not the main focus of our paper.
>
> **How does ergodic risk measure behave and adapt in highly non-stationary or non-ergodic environments where assumptions do not hold?**
>
> By definition (and as shown empirically in our work), unlike static and nested risk measures, ergodic risk measures are well-defined in non-stationary environments.
>
> When it comes to the ergodicity assumptions there is a distinction that needs to be made: In our paper we propose two different items: a class of risk measures (ergodic risk measures) and a specific RL objective that we show is an instance of an ergodic risk measure (Eq 2 in submitted draft, Eq 5 in updated draft). The ergodicity assumptions apply to the RL objective (not the encompassing class of ergodic risk measures).
>
> Now, what happens to the RL objective when the ergodicity assumptions do not hold? In the context of our paper, it means that we cannot *prove* that the objective satisfies the definition of an ergodic risk measure (we use the ergodicity assumption to show that a particular property is satisfied), however in terms of the empirical behaviour, a lack of ergodicity assumptions, roughly speaking, just means that we cannot *guarantee* that a unique optimal solution exists.
>
> How can it be extended using function approximation or deep RL architectures?
> The class of ergodic risk measures are well defined with function approximation. In terms of the aforementioned RL objective, this remains an area of active research (even risk-neutral average-reward RL with function approximation is relatively understudied).
>
> We again thank the reviewer for their review of our work. We are happy to continue discussing with the reviewer on any additional comments that the reviewer has.
>
> **References:**
>
> [1] Abel et al. 2023. “A Definition of Continual Reinforcement Learning.”
>
> [2] Abel et al. 2025. “Plasticity as the Mirror of Empowerment.”
>
> [3] Rojas et al. 2025 “Burning RED: Unlocking Subtask-Driven Reinforcement Learning and Risk-Awareness in Average-Reward Markov Decision Processes.”
>
> [4] Xia et al. 2023 “Risk‐sensitive Markov Decision Processes with Long‐run CVaR Criterion”

---

### Author Response · Authors · 2025-11-20
**Overall Rebuttal**

We want to thank all the reviewers for their insightful reviews of our work. In this overall response, we summarize the changes made to the paper based on the reviewers’ comments (the updated draft is now available), as well as address two common concerns brought up by multiple reviewers. We address more specific concerns and comments under each review.

**Summary of Changes**:

Based on a suggestion from one of the reviewers, instead of proposing our own definition of plasticity for a risk measure, we now utilize the definition of plasticity of an agent (and subsequent continual RL framework) from [1]. More specifically, we generalize the framework from [1] to the risk-aware case, and use the resulting framework to propose a set of axioms that establish the conditions necessary for risk-awareness in the continual setting. These axioms are motivated by limitations of the agent itself, rather than the risk measure.

Then, using these axioms, we follow a similar process as with the submitted draft to show that static and nested risk measures are not compatible with the axioms and that ergodic risk measures are.

We now address the two common concerns:

**Ergodicity Assumptions and Continual RL**:

Most of the reviewers argued that the ergodicity assumptions are too strong for the continual setting. In this regard, we have a few comments:

First, it is worth clarifying that in this work we propose a class of risk measures which we call ergodic risk measures. However, the ergodicity assumptions are not used to define this class of risk measures. Rather, the assumptions are used to show that the risk-aware version of the standard average-reward objective satisfies the definition of an ergodic risk measure, if the assumptions hold. Hence, the assumptions apply to the specific (average-reward-based) objective used, but not necessarily to the entire class of proposed ergodic risk measures.

Next, we note that, in our work, the risk-aware average-reward objective was introduced as the objective of an infinite sequence of average-reward MDPs. In this regard, we note that the ergodicity assumptions need only apply independently for each MDP, rather than holistically for the entire sequence of MDPs. For example, Assumption 4.3 (in the updated draft) only requires that each state in a given MDP, is accessible from every other state in that same MDP, but not the states of other MDPs. We acknowledge that this was not made clear in the submitted draft, and have made this clarification in the updated draft.

Finally, we disagree that ergodicity is incompatible with continual RL. To the contrary, having the ability to (eventually) reach any state in a given MDP is in alignment with continual RL. Consider the contrary, if the ergodicity assumptions do not hold, then the agent could get ‘stuck’ in one part of the MDP indefinitely, which would not be ideal in a continual RL setting.

**Ergodic Risk Measures vs Prior Works in Average-Reward RL:**

We received a few questions along the lines of: “You say that risk measure theory from non-continual RL does not work in the continual setting, but then use a prior algorithm from a non-continual setting [2] as an example of the proposed class of ergodic risk measure. How does that make sense?”

Essentially, there is a theory-practice gap that this paper addresses. In particular, there have been prior works that have explored the risk-aware version of the average-reward objective (e.g. [2, 3]) in non-continual settings. However, none of these prior works formalized this objective in the context of risk measure theory. Accordingly, until this work, risk measure theory in non-continual RL consisted only of static and nested risk measures. As such, not only does this work propose a risk measure that is compatible with continual learning, but it also formalizes the risk-aware average-reward objective in the context of measure theory, thereby closing a theory-practice gap that had existed until now. We have clarified this in the updated draft of the paper as Remark 4.1.

We again thank the reviewers and look forward to further discussions.

**References:**

[1] Abel et al. 2025. “Plasticity as the Mirror of Empowerment.”

[2] Rojas et al. 2025 “Burning RED: Unlocking Subtask-Driven Reinforcement Learning and Risk-Awareness in Average-Reward Markov Decision Processes.”

[3] Xia et al. 2023 “Risk‐sensitive Markov Decision Processes with Long‐run CVaR Criterion”

---

### Author Response · Authors · 2025-12-01
**Withdrawal Statement**

Dear Area Chair and Reviewers,

Given the recent developments in the review process, namely that the reviewers are no longer able to provide feedback on the updated draft of the paper, we do not see a way forward with our paper in this current review cycle, and have chosen to withdraw it.

We want to express our deepest gratitude to the reviewers for their willingness to consider both the submitted and updated drafts of our paper, and we are disappointed to not be able to continue what had been very insightful conversations that undoubtedly improved the quality of our paper.

Best regards,

Authors

---

### Note · Authors · 2025-12-02

I have read and agree with the venue's withdrawal policy on behalf of myself and my co-authors.